# Nutritional Support in Head and Neck Radiotherapy Patients Considering HPV Status

**DOI:** 10.3390/nu13010057

**Published:** 2020-12-27

**Authors:** Adam Brewczyński, Beata Jabłońska, Sławomir Mrowiec, Krzysztof Składowski, Tomasz Rutkowski

**Affiliations:** 1I Radiation and Clinical Oncology Department of Maria Skłodowska-Curie National Research Institute of Oncology, 44-102 Gliwice Branch, Poland; Adam.Brewczynski@io.gliwice.pl (A.B.); Krzysztof.Skladowski@io.gliwice.pl (K.S.); Tomasz.Rutkowski@io.gliwice.pl (T.R.); 2Department of Digestive Tract Surgery, Medical University of Silesia, 40-752 Katowice, Poland; mrowasm@poczta.onet.pl

**Keywords:** nutritional support, head and neck cancer, oropharyngeal cancer, human papillomavirus, radiotherapy

## Abstract

Malnutrition is a common problem in patients with head and neck cancer (HNC), including oropharyngeal cancer (OPC). It is caused by insufficient food intake due to dysphagia, odynophagia, and a lack of appetite caused by the tumor. It is also secondary to the oncological treatment of the basic disease, such as radiotherapy (RT) and chemoradiotherapy (CRT), as a consequence of mucositis with the dry mouth, loss of taste, and dysphagia. The severe dysphagia leads to a definitive total impossibility of eating through the mouth in 20–30% of patients. These patients usually require enteral nutritional support. Feeding tubes are a commonly used nutritional intervention during radiotherapy, most frequently percutaneous gastrostomy tube. Recently, a novel HPV-related type of OPC has been described. Patients with HPV-associated OPC are different from the HPV− ones. Typical HPV− OPC is associated with smoking and alcohol abuse. Patients with HPV+ OPC are younger and healthy (without comorbidities) at diagnosis compared to HPV− ones. Patients with OPC are at high nutritional risk, and therefore, they require nutritional support in order to improve the treatment results and quality of life. Some authors noted the high incidence of critical weight loss (CWL) in patients with HPV-related OPC. Other authors have observed the increased acute toxicities during oncological treatment in HPV+ OPC patients compared to HPV− ones. The aim of this paper is to review and discuss the indications for nutritional support and the kinds of nutrition, including immunonutrition (IN), in HNC, particularly OPC patients, undergoing RT/CRT, considering HPV status.

## 1. Introduction

Head and neck cancer (HNC) includes cancer of the oral cavity, pharynx, larynx, hypopharynx, and paranasal sinus [1]. Some risk factors have been described, including smoking, alcohol abuse, and human papillomavirus (HPV) infection [1,2]. HPV plays a role in the development of a subset of HNCs and, notably, oropharyngeal cancer (OPC) [3]. The role of HPV is not so clear in non-oropharyngeal carcinomas (non-OPCs), but some reports suggest a possible association between HPV infection and nasopharyngeal cancers [3,4,5]. HPV-induced OPC has different biological behavior and a better prognosis compared to non-HPV-induced OPC, and the eighth-edition Tumor-Node-Metastasis (TNM) classification now separates these two types [3,6,7].

Malnutrition is a very important problem in cancer patients. It has been reported in 15–80% of cancer patients [8,9]. The patients with HNC are particularly at risk of malnutrition due to the basic disease and complications of the oncological treatment such as radiotherapy (RT) or concurrent chemoradiotherapy (CRT). According to the literature, malnutrition is noted in 30–50% of HNC patients, especially located in the oropharynx and the hypopharynx [1]. It is caused by insufficient food intake due to dysphagia, odynophagia, and a lack of appetite caused by the tumor. It is also secondary to RT/CRT as a consequence of mucositis with the dry mouth, loss of taste, dysphagia, xerostomia, nausea, vomiting [1,10]. According to some authors, in up to 80% of HNC patients undergoing RT, significant weight loss during the treatment period is observed [11,12,13,14,15]. Severe dysphagia is a serious challenge in OPC patients because, in 20–30% of patients, it leads to the definitive total impossibility of eating through the mouth [10]. Therefore, CRT is associated with additional deterioration of the nutritional status (NS) in HNC patients. It has been shown that during RT or CRT, in 55% of patients, an additional weight loss (10% or more) had been observed [16,17,18]. It is also known that deterioration of NS is associated with increased CRT toxicity, increased risk of infections, which leads to prolonged treatment duration (due to treatment interruptions), poor clinical outcome, increased morbidity and mortality, and decreased quality of life (QoL) [1,11,16,19,20].

Therefore, it is obvious that nutritional support in order to improve NS in HNC patients is very important and necessary. Some questions should be answered. What is the best time for the beginning of the nutritional support in patients (before, during, or after CRT)? In which patients should nutritional support be started (malnourished or well-nourished)? What kinds and routes of nutritional support can be used? Is there any difference in nutritional status, nutritional deterioration, and nutritional support required depending on HPV status?

The aim of this paper is to answer mentioned above questions based on the review of the recent studies regarding nutritional support in HNC patients, particularly OPC patients, undergoing RT or CRT. The various types of nutritional support, such as nutritional counseling, oral supplements, enteral feeding, and parenteral nutrition, are discussed in this paper.

## 2. Dysphagia in HNC Patients: Reasons and Treatment

It should be noted that in HNC patients, dysphagia can appear before any oncological treatment and (secondary to the basic disease) and during or after oncological treatment (surgery, chemoradiotherapy) as the result of the treatment toxicity.

Dysphagia, defined as the difficulty in swallowing liquids, food, or medication, can occur during the oropharyngeal or the esophageal phase of swallowing. Swallowing dysfunction has been reported in 30–50% of non-surgically treated HNC. Before radiotherapy, dysphagia is caused by obstruction by the tumor volume or infiltration of structures involved with swallowing. In surgical patients, resection of structures necessary for normal deglutition leads to swallowing dysfunction. In patients receiving radiotherapy, dysphagia is secondary to injury of neural and soft tissues. RT-induced swallowing dysfunction may occur both during treatment and as a late effect of therapy. Acute dysphagia is generally associated with soft tissue inflammation, edema, pain, mucous production, and xerostomia. After radiation therapy has been completed, soft tissues are able to heal. In some patients, the healing process results in soft tissue fibrosis, lymphedema, scar tissue formation, and neurological impairment. This also may lead to swallowing dysfunction [21]. 

At the beginning of treatment, preventive swallowing dysfunction evaluation is used by nutritionists and deglutologists. Usually, a deglutologist identifies swallowing abnormalities, prescribes additional testing (clinical/radiological tests) in order to assess inhalation/aspiration risks, and develops an appropriate treatment plan (correction of swallowing mechanisms through patient education and exercises). Two types of exercises are recommended for patients with dysphagia. They can be performed at the beginning, during, and after treatment: indirect (exercises to strengthen swallowing muscles) and direct (postural exercises to be performed while swallowing). The aim of swallowing exercises is to increase the range of movement of the tongue, lips, and jaw. Exercising swallowing muscles improves and/or maintains the possibility of swallowing. The nutritional support involving artificial nutrition is the next step of dysphagia treatment. First of all, it includes dietary counseling, oral nutritional supplements, and enteral nutrition. If enteral nutrition is introduced, patients should be encouraged to continue to swallow and to wean from artificial nutrition as quickly and safely as is feasible, regardless of the nutrition method (e.g., nasogastric tube, percutaneous endoscopic gastrostomy (PEG), and parenteral nutrition) [21]. Depending on timing, dysphagia is divided into early (onset less than six months), interval (onset from six months to five–ten years), and late (onset more than six months). The pathogenesis of each above-mentioned dysphagia type is different [22]. The reasons for dysphagia in HNC patients are summarized in Table 1.

## 3. Methods of the Literature Research

We have reviewed PubMed and Web of Science databases. The search terms and mesh heading were as follows: “head and neck cancer”, or “oropharyngeal cancer”, and “HPV” or “nutritional support” or “nutritional intervention” or “nutrition” or “nutritional counseling” or “nutritional supplements” or “enteral nutrition” or “enteral feeding” or “nasogastric tube” or “feeding tube” or “gastrostomy” or “percutaneous endoscopic gastrostomy” or “prophylactic gastrostomy” or “reactive gastrostomy” or “parenteral nutrition” or “immunonutrition”. Selected articles related to the topic of our paper were read, analyzed, discussed, and cited. Full-text articles published in English were included in our review if they met the following criteria: Population: Head and neck cancer patients receiving radiotherapy or chemoradiotherapy, including HPV-related cancer patients.Interventions: nutritional counseling, oral nutritional supplements, enteral feeding including prophylactic vs. reactive nutrition, and nasogastric tube vs. percutaneous endoscopic gastrostomy, immunonutrition.Outcomes of interest: Optimal timing of nutritional support; dietary counseling (timing and indications), oral nutritional supplements (timing and indications); comparison of prophylactic and reactive enteral feeding; comparison of a nasogastric tube and percutaneous endoscopic gastrostomy, including following parameters: difference in body weight, rates of treatment interruption, nutrition-related hospital admission, and tube-related complications; the influence of HPV status on malnutrition and nutritional support; use of immunonutrition in HNC patients undergoing CRT.

The strategy of the literature review is presented in Table 2. The search results are presented in Table 3.

## 4. European Society for Clinical Nutrition and Metabolism (ESPEN) Guidelines on Enteral Nutrition: Non-Surgical Oncology

The European Society for Clinical Nutrition and Metabolism (ESPEN) guidelines on enteral nutrition (EN): non-surgical oncology were published in 2006 [23]. These guidelines are intended to give evidence-based recommendations for the use of oral nutritional supplements (ONS) and tube feeding (TF) in cancer patients. They were developed by an interdisciplinary expert group in accordance with officially accepted standards, are based on all relevant publications since 1985. We decided to present ESPEN recommendations regarding oral and enteral nutrition in cancer patients receiving RT or CRT because oral and enteral nutritional support is the most common in HNC patients. When a standard diet is not sufficient and or patients have problems considering eating, nutritional support involves the most physiological route at the beginning, starting from nutritional counseling, subsequently ONS, and then EN using a nasogastric tube (NGT) or gastrostomy. In the 2.2. paragraph, the question “when should EN be started?” is answered here. According to recommendations, nutritional therapy should be started if undernutrition already exists or if it is anticipated that the patient will not be able to eat for more than seven days. EN should be started if an inadequate food intake (<60% of estimated energy expenditure) is anticipated for more than ten days (grade of recommendation: C) [23].

In the 2.5. paragraph, the question “does supplementation with omega-3-fatty acids have a beneficial effect in cancer patients?” is answered here. According to experts, randomized clinical trial evidence is contradictory/controversial; therefore, it is not possible to reach any firm conclusion regarding the improvement of NS in patients receiving omega-3 fatty acids (grade of recommendation: C) [23].

In the 3.2. paragraph, authors recommended the use of intensive dietary counseling and ONS in order to increase dietary intake (grade of recommendation: A) and to prevent therapy-associated weight loss and interruption of the treatment in patients HNC undergoing RT (grade of recommendation: A). The authors recommended the use of tube feeding in patients with obstructing HNC coexisting with swallowing (grade of recommendation: C) [23]. TF is also suggested if severe local mucositis is expected, which might coexist with swallowing, e.g., in intensive RT or CRT regimens, including radiation of throat (grade of recommendation: C) [23]. TF can either be delivered via transnasal or percutaneous routes. Because of radiation-induced oral and esophageal mucositis, a percutaneous endoscopic gastrostomy (PEG) may be preferred (grade of recommendation: C) [23].

According to ESPEN recommendations, nutritional support involving at the beginning the most physiological routes, starting from nutritional counseling, subsequently ONS, in the next step EN using NGT or gastrostomy, is needed in patients with HNC undergoing RT/CRT, in whom the standard diet is not sufficient. Because of radiation-induced oral and esophageal mucositis, a percutaneous endoscopic gastrostomy (PEG) may be preferred in these patients.

## 5. ESPEN Guidelines on Parenteral Nutrition: Non-Surgical Oncology

The ESPEN guidelines on parenteral nutrition (PN) in non-surgical oncology were published in 2009. PN offers the possibility of increasing or ensuring nutrient intake in patients in whom normal food intake is inadequate, and EN is not feasible, is contraindicated, or is not accepted by the patient [24]. These guidelines are intended to provide evidence-based recommendations for the use of parenteral nutrition in cancer patients. They were developed by an interdisciplinary expert group in accordance with accepted standards, are based on the most relevant publications of the last 30 years, and share many of the conclusions of the ESPEN guidelines on enteral nutrition in oncology. Many indications for PN are similar to indications for EN (weight loss or reduction in food intake for more than seven–ten days), but only patients who can not be fed orally or enterally are candidates to receive PN. It is emphasized that in non-surgical well-nourished oncologic patients, routine parenteral nutrition is not recommended because it has proved to offer no advantage and is associated with increased morbidity [24].

According to the 2.2. recommendation, nutritional support should be started if a patient is undernourished or if it is anticipated that the patient will be unable to eat for more than seven days. It should also be started if inadequate food intake (<60% of estimated energy expenditure) is anticipated for more than ten days (grade of recommendation: C) [24]. If in such cases, nutritional support can not be delivered through the enteral route, it has to be delivered by a vein. A “supplemental” PN should substitute the difference between the actual oral/enteral intake and the estimated requirements (grade of recommendation: C) [24]. There is no rationale for the administration of PN if the nutrients intake by the oral or enteral route is adequate, and for these reasons, PN should not be given in such conditions (grade of recommendation: A) [24].

In HNC patients, the alimentary tract below the oral cavity, pharynx, and larynx is commonly non-impaired; therefore, PN is used very rarely in this patient group.

According to ESPEN guidelines, PN is recommended only for patients who can not be fed orally or enterally. In these patients, it has to be delivered by a vein. PN is also can be used additionally with EN as “supplemental” PN if the energy and protein requirements cannot be covered by the enteral route. “Supplemental” PN should substitute the difference between the actual oral/enteral intake and the estimated requirements.

## 6. The Optimal Timing of Nutritional Support in HNC Patients Undergoing CRT

Recently (September 2020), Hu et al. [25] published an article regarding the impact of early nutritional counseling in HNC patients with normal NS. The authors analyzed 243 HNC patients with normal NS before treatment undergoing concurrent CRT at three medical centers. All patients were retrospectively divided into three groups: (1) the early (≤2 weeks, *n* = 105, 43.2%), (2) late (>2 weeks, *n* = 102, 42.0%), and (3) no nutritional counseling groups (*n* = 36, 14.8%) according to the time interval between the date of CRT initiation and the first date of nutritional counseling. The 1-year overall survival rates were 95.0% (1), 87.5% (2), and 81.3% (3) (*p* = 0.035), respectively. The median body weight changes at the end of CRT were—4.8% (1), 5.6% (2), and—8.6% (3), respectively. The early termination of chemotherapy rates and the incompletion rates of planned RT were 1.9% and 1.9%, 2.9%, and 2.0%, 13.9%, and 19.4% in patients in the early and late groups, respectively. This study showed that early nutritional counseling was associated with a significantly increased overall survival, decreased loss of weight, and decreased treatment interruption. Therefore, early nutritional counseling is necessary for HNC patients [25].

In 2020, Jovanovic et al. [26] investigated the natural history of weight and swallowing outcomes in patients with and without feeding tube (FT) insertion. The authors analyzed retrospectively 122 OPC patients undergoing CRT at a single regional cancer center between January 2013 and December 2015. In 38 (31.1%) patients FT was introduced (FT group). The significant weight loss at 3 and 6 months in patients with and without FT and at 12 months in patients without a FT (NFT group) was reported. The significant decrease of swallowing-related QoL at 3 and 6 months was noted only in patients without FT. The authors concluded that clinically relevant loss of weight and decreased swallowing-related QoL in the first-year following treatment was noted in OPC patients regardless of FT insertion. Therefore, nutritional assessment and nutritional support are very important in OPC patients [26].

In 2010, Paccagenella et al. [27] assessed the impact of early nutritional support on the clinical outcome of HNC patients undergoing CRT. The authors retrospectively analyzed 33 HNC patients receiving early nutritional intervention (nutrition intervention group, NG) before CRT and 33 patients who underwent CRT without a specifically designed early nutritional support (control group, CG). These two patients groups were compared. The weight loss during CRT was significantly lower in NG patients compared to CG patients (−4.6% ± 4.1% vs. −8.1% ± 4.8% of pre-treatment weight, *p* < 0.01, after treatment). In the NG, fewer RT interruptions (>5 days) for toxicity (30.3% vs. 63.6%, *p* < 0.01) were reported. The mean number of days of RT delayed due to toxicity was 4.4 ± 5.2 in NG vs. 7.6 ± 6.5 in CG (*p* < 0.05). There were fewer patients who had an unplanned hospitalization in the NG compared to the CG (16.1% vs. 41.4%, *p* = 0.03). In the NG, symptoms influencing the NS were noted early and were present in almost all patients at chemotherapy completion. Additionally, 60.6% of NG patients needed tube feeding. The authors concluded that early nutrition support in HNC patients undergoing CRT was associated with improved tolerance of treatment and fewer admissions to hospital. This result suggests that nutritional intervention must be initiated before CRT, and it needs to be continued after treatment completion [27].

Regarding the optimal timing of nutritional support in HNC patients, the problem of so-called PEG dependency should be discussed. It has been suggested that the routine use of prophylactic PEG for nutritional support during radical CRT for HNC can lead to PEG dependency. In 2020, Fraser et al. [28] determined the rates of gastrostomy dependency at the Calvary Mater Newcastle (CMN), where PEGs were routinely used and identified potentially modifiable risk factors. In this study, medical records of 250 HNC patients with a prophylactic PEG, receiving radical CRT, between 2000 and 2015 were analyzed. Overall, eight patients (3%) were unable to wean. At 12 months following treatment, 16 patients (6%) still required PEG tubes for feeding. A greater tumor (T) advance (T4 or synchronous head and neck tumors) and the number of days Nil By Mouth (NBM) remained as significant independent risk factors for PEG dependency at 12 months. The authors concluded that the routine use of prophylactic PEG had not been associated with significant rates of PEG dependency at the CMN. Seeing a speech pathologist during treatment and intensity-modulated radiation therapy (IMRT) may decrease time NBM, which was identified as a potentially modifiable risk factor for PEG dependency [28].

Based on the literature, we recommend starting nutritional support in HNC patients before RT/CRT, continue it during the treatment, and maintain nutrition for 12 months following RT/CRT.

## 7. The Nutritional Counseling and Oral Nutritional Supplements in HNC Patients Undergoing CRT

Vlooswijk et al. [29] analyzed dietary counseling, nutritional support, body weight, and toxicity during and after treatment and investigated the effect of pretreatment body mass index (BMI) on survival in OPC patients. This retrospective study involved 276 OPC patients receiving RT. Endpoints were dietary consultations, weight loss, toxicity, overall survival, and disease-free survival. A retrospective chart review was done in 276 OPC patients undergoing RT. Endpoints were dietary consultations, weight loss, toxicity, overall survival, and disease-free survival. Almost all OPC patients received dietary counseling (94%) and nutritional support (99%). Dietary counseling decreased sharply shortly after treatment to 38% at one year after treatment. Overall weight loss increased during the first year of follow-up and ranged from 3% at the start of RT until 11% at one year after RT. Overall survival was significantly longer in patients with a BMI above average (*p* = 0.01). Acute dysphagia (*p* = 0.001), mucositis (*p* = 0.000) and a toxicity grade of three (*p* = 0.002) were significantly more prevalent in patients who had lost 10% or more of their body weight. This study showed that patients continued to lose body weight during and until one year after RT, despite nutrition support and frequent dietary counseling. A BMI above-average appears to increase survival time. This study showed that nutritional counseling in OPC patients is necessary for a long time after RT [29].

Kabarriti et al. [30] analyzed nutritional factors and compliance with dietary recommendations for associations with overall survival (OS) and progression-free survival (PFS) in patients receiving definitive RT for laryngeal and oropharyngeal cancers. The authors identified 352 patients with non-metastatic laryngeal (146) and oropharyngeal (206) cancer treated with definitive RT between 2004 and 2013. The median follow-up was 22.86 months. The rates for OS were 91, 86, and 73% at years one, two, and five, respectively. The majority (85.8%) of patients met with a dietitian during their course of RT, and 62.6% of these patients were compliant with the nutritional program. Compliance with the nutritional program resulted in 27% (HR 0.73, 95% confidence interval (CI) 0.43–1.26) protection from death (not significant) and 31% (HR 0.69, 95% CI 0.50–0.94) significant protection from disease progression. Higher pretreatment BMI was associated with a lower risk of death (HR 0.94, 95% CI 0.90–0.99), and disease progression (HR 0.96, 95% CI 0.93–0.99). The authors concluded that laryngeal and oropharyngeal cancer patients receiving definitive RT with regular dietetic counseling and contact had improved outcomes [30].

Valentini et al. [31] analyzed patients undergoing CRT for HNC who received nutritional counseling by the dietitian within the first four days of RT and weekly for the course of RT (approximately six weeks). A weekly supply of oral nutrition supplements 1560 kJ (373 kcal) per 100 g for up to three months was provided to all patients. Twenty-one patients completed CRT. Mucositis grade 3 (G3) was noted in seven (33.3%) patients, whereas mucositis G4 was absent. Dysphagia was present before the start of treatment in four patients. In the remaining 17 patients, dysphagia G3 developed during/at the end of treatment in five cases. The percentage of patients interrupting oncologic treatment was 28% for ≥six days, 28% for three–five days, and 44% for zero–two days. Mucositis G3 frequency was lower in patients with a baseline BMI ≥ 25 kg/m^2^ (two out of 12; 16.6%) compared to patients with BMI < 25 (five out of nine; 55.5%) (*p* = 0.161) and in patients with a baseline mid-arm circumference > 30 cm compared to patients with a mid-arm circumference 28.1–30 cm and <28 cm, and higher in patients with a higher weight loss and a greater reduction of serum albumin and mid-arm circumference. The authors concluded that nutritional counseling and ONS are associated with relatively low CRT-related toxicity and with a mild deterioration of nutritional parameters [31].

Recently (June 2020), Mello et al. [32] published a very interesting meta-analysis on nutritional support, including nutritional counseling and ONS. The analysis included 15 trials, of which five were ongoing or unpublished, providing evidence in four comparisons. The authors found very low to low certainty evidence for the impact of ONS on mortality, treatment tolerance, QoL, functional status, and adverse effects. In comparison to nutritional counseling alone, nutritional counseling plus oral ONS probably increased body weight slightly. The authors also found adverse events relating to supplements use such as nausea, vomiting, and feeling of fullness. In the authors’ opinion, further investigations are needed due to possible increases in mortality, treatment tolerance, and quality of life besides a possible decrease in functional status due to ONS. It is important to remember about possible adverse effects of the use of ONS. They should not be overlooked [32].

Bicakli et al. [33] assessed the effects of compliance of patients with individual dietary counseling on body composition parameters in 69 HNC patients undergoing RT. All patients received nutritional counseling, and the majority of them (94.6%) received ONS during RT or CRT. If a patient consumed ≥75% of the recommended energy and protein intake via ONS and regular food, he/she was considered to be “compliant” (*n* = 18), while those who failed to meet these criteria were considered to be “non-compliant” (*n* = 30). BMI, weight, fat percentage, fat mass, fat-free mass, and muscle mass did not decrease significantly over time in compliant patients, but in noncompliant patients, all of these parameters decreased significantly from baseline compared to the end of treatment (*p* < 0.001). There was no significant difference in the hand-grip strength between the two groups at baseline and over time in each group. In a retrospective analysis, heavy mucositis was less frequently noted in compliant than non-compliant patients (11.1% versus 88.9%, respectively) (*p* < 0.009). The authors concluded that body composition parameters were better in HNC patients considered as compliant with nutritional counseling than non-compliant ones during the RT period [33].

Van den Berg et al. [34] assessed the value of individually adjusted counseling by a dietitian compared to standard nutritional care (SC). A prospective study, conducted between 2005 and 2007, compared individual dietary counseling (IDC), optimal energy, and protein requirement, to SC by an oncology nurse (standard nutritional counseling). Endpoints were weight loss, BMI, and malnutrition (5% weight loss/month) before, during, and after the treatment. Thirty-eight patients were included evenly distributed over two groups. A significant decrease in weight loss was found two months after the treatment (*p* = 0.03) for IDC compared to SC. Malnutrition in patients with IDC decreased over time, while malnutrition increased in patients with SC (*p* = 0.02). The authors concluded that early and intensive individualized dietary counseling by a dietitian resulted in clinically relevant effects in terms of decreasing weight loss and malnutrition compared with SC in HNC patients undergoing radiotherapy [34].

Ferreira et al. [35] investigated the association between treatment time points and oral nutritional supplementation (ONS) on dietary intake to estimate the frequency of energy and nutrient inadequacy and also to evaluate body weight changes (BWC). Dietary intake data of 65 patients were obtained from 24-h dietary recalls, and prevalence of inadequacy was calculated before or at the beginning (T0), in the middle (T1), and at the end of treatment (T2). In this study, the decreased energy and macronutrient intake at the middle of treatment and the increased micronutrient intake due to ONS use were noted in HNC patients. Despite this, the prevalence of inadequate energy and nutrient intake, particularly for calcium, magnesium, and vitamin B6, was high at all-time points even with ONS use but proved worse for those who did not use ONS. In particular, calcium, magnesium, and vitamin B6 showed an almost 100% probability of inadequacy for those who did not use ONS. The overweight patients suffered a higher weight accumulated deficit compared to other BMI categories. Patients receiving ONS showed a lower weight loss. Therefore, the authors strongly recommended initiating nutritional counseling from diagnosis to optimize macronutrient intake in conjunction with prophylactic ONS prescription to adjust micronutrient intake and minimize the weight loss, making it possible to prevent worse prognosis and NS [35].

Cereda et al. [36] assessed the benefit of ONS in addition to nutritional counseling in HNC patients undergoing RT. In a single-center, randomized, pragmatic, parallel-group controlled trial, 159 newly diagnosed HNC patients suitable for RT regardless of previous surgery and induction chemotherapy were randomly assigned to nutritional counseling in combination with ONS *(n* = 78) or without ONS (*n* = 81) from the start of RT and continuing for up to three months after its end. The primary endpoint was the change in body weight at the end of RT. Secondary endpoints included changes in protein-calorie intake, muscle strength, phase angle and quality of life, and anti-cancer treatment tolerance. In patients with the primary endpoint assessed, counseling plus ONS (*n* = 67) was associated with smaller weight loss compared to nutritional counseling alone (*n* = 69; mean difference, 1.6 kg (95%CI, 0.5–2.7); *p* = 0.006). Additionally, in patients with ONS supplementation, higher protein-calorie intake and improvement in the quality of life over time were observed (*p* < 0.001 for all). The use of ONS reduced the need for changes in scheduled oncological treatment (i.e., for RT and/or systemic treatment dose reduction or complete suspension, HR = 0.40 (95%CI, 0.18–0.91), *p* = 0.029). The authors concluded that in HNC patients undergoing RT or CRT and receiving nutritional counseling, the use of ONS resulted in better weight maintenance, increased protein-calorie intake, improved quality of life, and was associated with better anti-cancer treatment tolerance [36].

Ravasco et al. [37] assessed the effect of dietary counseling or oral supplements on outcome in 75 HNC patients undergoing RT. The patients were randomized to the following groups: group 1 (*n* = 25), patients who received dietary counseling with regular foods; group 2 (*n* = 25), patients who maintained usual diet plus supplements; and group 3 (*n* = 25), patients who maintained intake ad lib. Nutritional intake and NS and QoL were assessed at baseline, at the end of RT, and at 3 months after RT. Energy intake after RT increased in both groups one and two (*p* ≤ 0.05). Protein intake also increased in both groups one and two (*p* ≤ 0.006). Both energy and protein intake decreased significantly in group three (*p* < 0.01). At three months following RT, group one maintained intakes, whereas groups two and three returned to baseline or below baseline levels. At three months after RT, the reduction of incidence/severity of grade one + two anorexia, nausea/vomiting, xerostomia, and dysgeusia were different: 90% of the patients improved in group one versus 67% in group two versus 51% in group three (*p* < 0.0001). After RT, QoL function scores improved (*p* < 0.003) proportionally with improved nutritional intake and status in group one/group two (*p* < 0.05) and worsened in group three (*p* < 0.05). At three months after RT, patients in group one maintained or improved overall QOL, whereas patients in groups two and three maintained or worsened overall QoL. The study showed that nutritional support positively influenced clinical outcomes in HNC patients undergoing RT [37].

Langius et al. [38] performed a systematic review to assess the impact of nutritional interventions on nutritional status, QoL, and mortality in HNC patients receiving RT or CRT. The authors analyzed 12 study reports describing ten different studies with 11 interventions. Four out of 10 studies examined the effects of individualized dietary counseling and showed significant benefits on nutritional status and QoL compared to no counseling or general nutritional advice by a nurse (*p* < 0.05) [38].

Nutritional support, using dietary counseling and ONS, should be implemented in HNC patients undergoing RT/CRT as early as possible as the most physiological method of nutrition of choice in patients, who can be fed orally. It improves outcome and QOL in this patient group.

## 8. The Enteral Nutrition in HNC Patients Undergoing CRT

Enteral nutrition (EN) is often required for nutrition support in HNC patients when oral intake is inadequate [39]. It may be used in order to prevent malnutrition and dehydration in patients undergoing CRT prior to the beginning of treatment (prophylactic or preventive nutrition) or in order to improve NS in already malnourished patients during oncological treatment (reactive nutrition). The patients may be fed enterally via nasogastric tube or PEG [16]. There are plenty of studies comparing the mentioned above nutritional timing types and routes in the world literature.

### 8.1. Prophylactic Versus Reactive Feeding Tube in HNC Patients Undergoing CRT

Vangelov et al. [39] retrospectively investigated the use of reactive feeding tubes (RFTs) in 131 patients with advanced OPC undergoing CRT. Predictors for RFT insertion were investigated. Weight loss during RT was compared between those with RFT versus prophylactic tubes (PFTs) versus no tube, and survival outcomes were evaluated. RFTs were significantly necessary in patients with bilateral neck node irradiation (*p* = 0.001) and CRT (*p* = 0.038). In patients with RFTs, significantly higher mean percentage weight loss during RT (9.5 ± 3.4%) (*p* < 0.001) than in patients with a PFT (5.2 ± 4.7%) and patients with no tube (5.4 ± 3.1%) was noted. Five-year survival rates in patients without a feeding tube and patients requiring a RFT were comparable. The authors concluded that RFT use was frequent in OPC patients, although difficult to predict. The study showed that CRT and bilateral neck node irradiation were associated with the need for the insertion of a feeding tube during RT. Also, in these patients, significantly higher weight loss was more frequently noted. Therefore, in the authors’ opinion, prophylactic tubes may need to be considered more often in this patient group [39].

Cady [40], based on the literature review, concluded that patients who required therapeutic PEG tube placement in response to significant weight loss during treatment suffered increased morbidity compared to patients receiving PEG tubes prophylactically. In the author’s opinion, various patient-, tumor-, and treatment-related risk factors should be taken into consideration in order to identify patients most likely to benefit from prophylactic PEG tube placement [40].

Lang et al. [41] retrospectively assessed PEG tube placement depending on body weight and BMI in 186 HNC patients undergoing RT. The authors also evaluated the course of weight change following PEG placement. Patients with dysphagia prior to treatment initiation and patients with a BMI < 18.5 kg/m^2^ needed PEG placement earlier during the treatment course. Low-grade toxicities related to PEG insertion were observed in 10.7% of patients, with the most frequently noted being peristomal pain and redness adjacent to the PEG tube insertion site. High-grade toxicities, including peritonitis and organ injury, were observed in 4.9% of patients [41]. The authors concluded that underweight patients and patients with preexisting dysphagia should be screened during RT for weight loss and decreased oral intake. The early PEG-tube placement should be considered for weight loss greater than 4.5% during the HNC treatment [41].

Brown et al. [42] published results of a prospective comparative cohort study in comparing prophylactic and reactive nutritional support in HNC patients. Patients were observed during routine clinical practice over two years. Patients were divided into two groups, according to protocol adherence as to whether they received prophylactic gastrostomy (PEG) per protocol recommendation (prophylactic PEG group, *n* = 69) or not (no PEG group, *n* = 61). The significantly less weight loss in the prophylactic PEG group (7.0% vs. 9.0%; *p* = 0.048) and more unplanned admissions in the no PEG group (82% vs. 75%; *p* = 0.029) were noted. In the no PEG group, 26 patients (43%) required a feeding tube or had ≥10% weight loss. The authors concluded that prophylactic gastrostomy improved nutrition outcomes and reduced unplanned hospital admissions [42]. It is interesting that the same author, in her previous study, did not note any beneficial effect of early nutritional support compared to standard care [43]. Patients were divided into groups: intervention or standard care. The intervention group started supplementary tube feeding immediately following tube placement. In this study, intervention did not influence on weight loss (10.9 ± 6.6% standard care vs. 10.8 ± 5.6% intervention, *p* = 0.930). No other differences were found for QoL or clinical outcomes. Also, any serious adverse events were not reported. The authors explained these results and concluded that the early intervention did not improve outcomes, but poor adherence to nutrition recommendations impacted potential outcomes [43].

Zhang et al. [44] compared effects of prophylactic percutaneous endoscopic gastrostomy (pPEG), reactive percutaneous endoscopic gastrostomy (rPEG), and nasogastric tube (NGT) in HNC patients undergoing CRT. Thirteen studies involving 1631 patients were included in this meta-analysis. The study showed that both pPEG and NGT were superior to rPEG in the treatment of weight loss. pPEG was associated with the least rate of treatment interruption and nutrition-related hospital admission among pPEG, rPEG, and NGT. The tube-related complications were comparable in all analyzed groups. The authors indicated pPEG as a better choice in malnutrition management in HNC patients undergoing CRT [44].

The different results were shown a study by Kramer et al. [45]. The authors analyzed 74 HNC patients undergoing RT with or without surgical resection. They received a PEG tube either before RT initiation (prophylactic) or after (reactive). Patients receiving reactive PEG tubes had them in place for fewer days than those placed prophylactically (227 vs. 139 days, *p* < 0.01). There was no difference in weight loss at two, six, or 12 months. Moreover, in this study, there was no difference in survival or disease control between the groups. In the authors’ opinion, reactive PEG tube placement may afford patients a shorter duration of usage without greater weight loss or poorer oncologic treatment results [45].

Willemsen et al. [46], in the interesting retrospective cohort study, developed a prediction model to identify patients who need prophylactic gastrostomy insertion, which was defined as the expected use of tube feeding for at least four weeks. This study involved 450 HNC patients undergoing CRT or bioradiation (BRT). The primary endpoint of this study was the use of tube feeding for at least four weeks during CRT/BRT or within 30 days after CRT/BRT completion. The four-week cut off point was based on the Dutch national dietary guidelines, recommending gastrostomy insertion as being superior to NGT when tube feeding is required for a period of four weeks or longer. According to the Dutch guideline on malnutrition, patients were initially recommended to use oral nutritional supplements or tube feeding in addition to oral nutrition when 50–75% of the calculated nutritional requirements were met. When oral intake was less than 50% of the calculated nutritional needs, without rapid improvement of oral intake, full tube feeding was used. Sixty-five percent (294/450 patients) required tube feeding for four weeks or longer. The authors’ final model includes the following predictors: BMI and adjusted diet at the start of CRT/BRT, percentage weight change at baseline, World Health Organization performance status, tumor subsite, TNM-classification, CRT/BRT, mean radiation dose on the contralateral submandibular and parotid gland. After internal validation, the model had good accuracy. The corrected Area Under the Curve (AUC) was 72.3% in discriminating HNC patients planned for CRT/BRT who would, versus would not, need TF for at least four weeks and thus would benefit from prophylactic gastrostomy insertion. This study is important and useful in order to avoid non-necessary PEG placement in order to avoid complications associated with gastrostomy insertion, such as PEG dependency [46].

This problem was also pointed out in the study by Fraser et al. [28]. The authors analyzed the risk of PEG dependency as a result of suggestion noticed that the routine use of prophylactic PEG tubes for nutrition support during radical CRT in HNC patients might result in PEG dependency. Following their analysis, the authors concluded that the routine use of prophylactic PEGs had not resulted in significant rates of PEG dependency at the Calvary Mater Newcastle (CMN). In the authors’ opinion, seeing a speech pathologist during treatment and intensity-modulated radiation therapy (IMRT) may decrease the number of days Nil By Mouth (NBM), which was identified as a potentially modifiable risk factor for PEG dependency [28].

Van der Linden et al. [47] analyzed 240 HNC patients undergoing CRT in order to create a protocol of qualification of HNC patients for insertion of PEG. In total, 195 patients used EN (via PEG or nasogastric tube). The study showed that the increased age (*p* = 0.01), nodal invasion (*p* = 0.02), reconstruction extent other than primary closure (*p* = 0.02), bilateral neck irradiation (*p* < 0.01), and an adapted intake consistency prior to CRT (*p* = 0.03) were significantly associated with PEG insertion. The authors concluded that nodal invasion and planned bilateral neck irradiation should be considered as indications for PEG insertion [47]. 

Sieron et al. [48] retrospectively analyzed the safety of prophylactic gastrostomy tube placement and usage in HNC patients. The study included HNC patients undergoing percutaneous endoscopic gastrostomy (PEG) or radiological percutaneous gastrostomy (RPG) tube placement prior to CRT. A total of 212 patients underwent prophylactic feeding tube placement (71% RPG, 27% PEG and 2% surgical jejunostomy). A total of 173 patients used their gastrostomy tubes for either total or supplemental nutrition support. Despite this, in 157 patients (74%), weight loss during CRT (mean weight loss = 8 kg) was reported. The severe complications rate (peritonitis/incorrect placement) was low and comparable in both groups (PEG 2.7% vs. RPG 3.4%). The authors concluded that although a very high proportion of patients used their PEG/RPG during CRT, the high loss of weight was noted. Serious complications of tube placement were rare [48].

Haussman et al. [49] analyzed 181 HNC patients using prophylactic PEG. The PEG was used in 91.7% of patients. One hundred and forty-nine patients (82.3%) used the PEG tube for total enteral nutrition, 17 patients (9.4%) for supplemental nutrition. Fifteen patients (8.3%) did not use the PEG tube. Peristomal wound infections were the most common complications (40.3%) in this study. A high Nutritional Risk Screening (NRS) score prior to tube insertion was found to be independently associated with PEG utilization. No significant weight changes were observed across the three patient subgroups. The authors concluded that although the overall PEG tube utilization rate was high, due to the infectious complications rate. However, the proper patient selection for PEG insertion is crucial in order to determine which patients benefit most from prophylactic PEG [49].

Yanni et al. [50] retrospectively analyzed 152 patients treated with surgery, RT, or chemotherapy for HNC. The patients were classified according to their gastrostomy status (41 prophylactic or non-prophylactic). Forty-one patients received prophylactic gastrostomy, whereas 111 patients had no nutritional support. Prophylactic gastrostomy placement was associated with a lower initial BMI, severe malnutrition, and with initial oral intake disorder. Patients who did not receive prophylactic gastrostomy had significantly worse outcomes such as hospital readmissions (*p* = 0.042), relative weight loss at six weeks (*p* < 0.0001), dysphagia, severe malnutrition, and a poor state of health (*p*= 0.001). Therefore, the authors concluded that prophylactic PEG showed advantages in terms of hospital readmissions, relative weight loss at six weeks, dysphagia, severe malnutrition, and poor health status [50].

McClelland et al. [51], in a systematic review, analyzed the optimal timing of PEG placement. The authors compared outcomes of patients with prophylactic PEG (pPEG) versus reactive PEG (rPEG). Twenty-two studies examining the role of PEG placement in CRT for aHNC were reviewed in this study. The authors noted that pPEG reduced the number of malnourished patients (defined as >10% of body weight), but average weight loss at various time points following treatment appeared similar to patients with rPEG. pPEG was also associated with improved quality of life at six months and greater long term PEG dependence. Based on this review study, the authors recommended inserting pPEG in those patients with the greatest risk of becoming malnourished during the course of treatment and using rPEG in remaining patients [51].

Strom et al. [52] analyzed retrospectively 297 OPC patients undergoing CRT in order to determine risk factors of reactive PEG placement. In this study, 128 patients did not receive a prophylactic PEG tube within ten days of CRT initiation. Fifteen of 128 patients (11.7%) required the reactive PEG insertion during or within three months of CRT. The median time for PEG tube removal was 3.3 months, and 14 of 15 patients had their PEG tube removed at the last follow-up analysis. Independent risk factors for PEG tube placement were as follows: accelerated irradiation fractionation (odds ratio (OR), 4.3; 95% CI, 1.1–16.5; *p* = 0.04), a tumor T classification ≥ three (OR, 3.5; 95% CI, 1.0–11.9; *p* = 0.04), a cumulative cisplatin dose ≥ 200 mg/m^2^ (OR, 6.7; 95% CI, 1.2–36.7; *p* = 0.03), and BMI < 25 (OR, 5.8; 95% CI, 1.4–23.9; *p* = 0.02) [52].

Silander et al. [53], in a prospective randomized study, analyzed 134 HNC patients divided into two groups: prophylactic PEG (study group) or clinical praxis (control group). Duration of hospitalization was comparable in both groups. After six months, QoL was significantly better, and the weight loss was significantly lower in the prophylactic PEG group. In this study, prophylactic PEG was associated with a significantly earlier start and longer use of enteral nutrition, fewer malnourished patients over time, and improved quality of life six months following CRT initiation [53].

Assenat et al. [54], in a retrospective study, analyzed 139 patients undergoing CRT. In this study, 78 (58%) patients did not receive prophylactic PEG feeding, and 61 (44%) patients received PEG feeding. Nutritional status before treatment was worse in the PEG group. In the PEG groups, the authors noted a significantly lower cumulative incidence of treatment interruption because of toxicity was compared to the non-PEG group (100 and 236 days of interruption, respectively, *p* = 0.03) and significantly shorter duration of hospitalization (*p* = 0.003) [54].

O’Shea et al. [55] analyzed the impact of smoking and alcohol consumption on the duration of gastrostomy tube (GT) use and dependence rates. This retrospective study involved 104 HNC patients undergoing RT/CRT. Prophylactic gastrostomy was inserted prior to treatment. The median duration of gastrostomy tube use was nine months. The actuarial GT persistence rate at one year was 35%. On univariate analysis, current smoking (hazard ratio (HR), 0.47; 95% CI, 0.27–0.81; *p* = 0.01) and current heavy alcohol consumption (HR, 0.55; 95% CI, 0.32–0.97; *p* = 0.04) were significant predictors of GT persistence. On multivariate analysis, only current smoking remained significant (HR, 0.53; 95% CI, 0.30–0.94; *p* = 0.03) [55].

According to most authors, prophylactic NGT/gastrostomy improved nutrition outcomes and reduced unplanned hospital admissions. However, the proper patient selection for PEG insertion is crucial in order to determine which patients benefit most from prophylactic PEG. Patients should be qualified for prophylactic or reactive NGT/gastrostomy individually, depending on the risk of malnutrition. There are some criteria scores in the literature in order to select patients who are the best candidates for prophylactic EN, such as the protocol from Van der Linden et al. [47], including nodal invasion and planned bilateral neck irradiation as indications for PEG insertion. Patients who will really benefit from prophylactic gastrostomy should be correctly selected. The proper selection of candidates for prophylactic EN is crucial in order to prevent malnutrition while not exposing them to complications related to unnecessary enteral access.

### 8.2. Nasogastric Tube Versus Percutaneous Endoscopic Gastrostomy in HNC Patients Undergoing CRT

Sadasivan et al. [56] compared the results of a nasogastric tube and PEG in HNC patients undergoing CRT. Patients were divided into two groups: a PEG group and a nasogastric tube (NGT) group. Each group involved 50 patients. Patients were assessed at weeks one and six and at six months. NGT patients had significant local site infection (64%), mainly vestibulitis, sinusitis, and epistaxis, whereas the PEG group had only 4% local site infection in the form of cellulitis and erythema (*p* < 0.001). There was no tube dislodgement in the PEG group, whereas in the NGT group, there was a 36% rate of tube dislodgement that required reinsertion (*p* < 0.001). Considering quality of life, the PEG group showed a statistically significant advantage over NGT group in all aspects (*p* < 0.001) according to the modified European Organization for Research and Treatment of Cancer Quality of Life Questionnaire-HandN35 (EORTC QLQ-HandN35). This study showed statistically significant superiority of PEG over NG tubes in terms of patient satisfaction, complications, and nutritional requirements [56].

Corry et al. [57] compared results of PEG and NGT in enteral nutrition in HNC patients undergoing CRT. This prospective study involved 32 PEG and 73 NGT patients. PEG patients presented significantly less weight loss at six weeks after treatment (median 0.8 kg gain vs. 3.7 kg loss, *p* < 0.001), but had a high insertion site infection rate (41%), the longer median duration of use (146 vs. 57 days, *p* < 0.001), and more grade three dysphagia in disease-free survivors at six months (25% vs. 8%, *p* = 0.07). Patient self-assessed general physical condition and overall QoL scores were comparable in both groups. Overall costs were significantly higher for PEG patients. The authors concluded that PEG tube use should be selective, not routine, in this patient population [57].

Mekhail et al. [58] also compared results of PEG and NGT in enteral nutrition in HNC patients undergoing CRT. The authors retrospectively analyzed 158 patients over the eight-year interval. Nasogastric tubes were inserted in 29 patients, and PEG tubes were placed in 62 patients. In PEG patients, dysphagia at three months (59% vs. 30%, respectively; *p* = 0.015) and at 6 months (30% vs. 8%, respectively; *p* = 0.029) was reported more frequently compared to NG patients. The median tube duration was significantly higher in PEG patients (28 weeks) compared to NG patients (eight weeks) (*p* < 0.001). Significantly more PEG (23%) needed pharyngoesophageal dilatation compared to 4% NGT patients (*p* = 0.022) [58].

Nugent et al. [12] conducted systematic searches for randomized controlled trials in order to compare different methods of enteral nutrition. Only randomized controlled trials were included in this analysis. Complications rates and patients satisfaction were comparable in both groups. The duration of PEG feeding was significantly longer than for the NG group (*p* = 0.0006). Moreover, the cost of PEG feeding was ten times greater than that of NG. In conclusion, the authors did not indicate the optimal method of enteral feeding [12].

Magne et al. [59] analyzed 90 HNC patients undergoing CRT, including 50 patients were with percutaneous fluoroscopic gastrostomy (PFG) and 40 patients with a nasogastric tube (NG). Mechanical failure was observed in 32/40 patients and in 7/50 of the gastrostomy group. In the PFG group, 80% of patients conserved nutrition after the end of the radiotherapy, and no patients in the NG group. In the PFG group, two patients presented with a wound infection, and in six patients, aspiration pneumonia was noted. In the NG group, in 21 patients, aspiration pneumonia, probably due to the NG tube (gastroesophageal reflux), was noted. The body weight and BMI at three weeks and at six weeks were comparable in both groups. Advantages were associated with PFG cosmesis, mobility, and QoL. The authors concluded that PFG was a safe and effective EN method for HNC patients undergoing CRT and offered important advantages over NG [59].

Williams et al. [60] retrospectively analyzed 104 OPC patients undergoing CRT, including 71 patients receiving prophylactic gastrostomy, 21 were managed with a NG tube as required, and 12 received a therapeutic gastrostomy. Patients with a prophylactic gastrostomy started EN significantly earlier (a median of 24 days following commencing RT), compared to a median of 41 days (*p* < 0.001) in the NG group. The median number of unplanned inpatient days were six, 14, and 7, respectively (*p* < 0.01 for prophylactic gastrostomy vs. NG patients). Mean percentage weight loss at the end of treatment (6.1% vs. 7.1% vs. 5.2%, respectively) and at six months following RT (11.7%, 14.3% and 8.9%) were comparable in all groups (*p* = 0.23). The median duration of enteral feeding was 181, 64, and 644 days, respectively (*p* < 0.01 for prophylactic gastrostomy vs. NG groups). Use of a prophylactic gastrostomy (*p* < 0.01) and higher T stage (*p* < 0.01) were significantly associated with increased duration of enteral feeding [60].

Ward et al. [61] analyzed 78 disease-free HPV-negative laryngopharynx cancer patients after CRT who had received a feeding tube. The five-year incidence of severe late dysphagia was 30.8% in the reactive nasogastric tube (R-NG) group (*n* = 36), 56.4% in the reactive percutaneous endoscopic gastrostomy (R-PEG) (*n* = 17; *p* = 0.193), and 60.9% in the proactive percutaneous endoscopic gastrostomy (P-PEG) (*n* = 25; *p* = 0.016) groups. On multivariate analysis, PEG feeding was independently associated with an increased rate of severe late dysphagia. The authors concluded that reactive nasogastric tube R-NG use during CRT was associated with less severe late dysphagia and is preferred over PEG [61].

Considering indications for one of two EN methods in HNC patients, Vlooswijk et al. [31], in the above-mentioned study, started nutritional support with dietary counseling. When the intake of energy and protein was not sufficient, a nasogastric tube was inserted. In these patients, PEG was introduced when the expected EN duration was more than three months. Comparison of two enteral feeding methods was not the issue of this study [31].

The reports on the optimal method of enteral nutrition are contradictory. Some authors have shown a statistically significant superiority of PEG over nasogastric tubes in terms of patient satisfaction, complications, and nutritional requirements. Additionally, the authors indicated the advantages of percutaneous gastrostomy, such as cosmesis, mobility, and QoL [56,59]. Other authors have not indicated an optimal method of enteral feeding. We recommend the nasogastric tube in patients receiving enteral nutrition in duration not longer than four weeks. In patients, whom the longer (more than four weeks) enteral feeding is planned, we recommend PEG insertion.

## 9. The Influence of HPV Status on Nutritional Status and Nutritional Intervention in HNC Patients Undergoing CRT

In 2018, Anderson et al. [62] presented a risk stratification model for feeding tube use in HNC patients undergoing intensity-modulated radiotherapy (IMRT). This study involved 139 patients treated with definitive IMRT (± concurrent chemotherapy) for head and neck mucosal cancers. The authors concluded that T classification ≥ three and level two lymphadenopathy could potentially stratify patients into four risk groups for developing severe dysphagia requiring feeding tube use. The following risk groups of prolonged FT use were stratified: (1) High risk (HRi)—T-classification ≥ three and level two Lymphadenopathy, (2) High-intermediate risk (HIRi)—T-classification ≥ three and No level two Lymphadenopathy, (3) Low-Intermediate Risk (LIRi)—T-classification < three and level two Lymphadenopathy, (4) Low risk (LRi)—T-classification < three and No level two Lymphadenopathy.

In the next study, Anderson et al. [63] analyzed 101 OPC patients undergoing RT. Patients were divided into high risk (HRi: T-classification ≥ three with level two Nodal disease), high-intermediate risk (HIRi: T-classification ≥ three without level two Nodes), and low-intermediate risk (LIRi: T-classification < three with level two Nodes) of prolonged feeding tube use. LIRi, HIRi, and HRi patients were defined as patients who have a median feeding tube (FT) use of ≥25% of their nutritional requirement for 75, 108, and 170 days, respectively. Weight-loss outcomes were reported according to their risk stratification level: high risk (T3 or T4 with level two lymphadenopathy), high-intermediate risk (T3 or T4 without level two lymphadenopathy), and low-intermediate risk groups (T0, T1 or T2 with level two lymphadenopathy). Overall, there was good adherence (87%) to their prophylactic feeding tube recommendations. The weight-loss outcomes were similar for patients with a FT (*n* = 87) compared to the whole cohort (*n* = 101). Interestingly, in the low-intermediate risk group, significantly higher loss weight compared to the high risk and high-intermediate risk groups was reported; 8.2% vs. 4.8% and 5.2%, respectively. Furthermore, in the low-intermediate risk group, the high rates of feeding tube placement (*n* = 36 received a tube out of the 42 recommended in this group) and a lower disease burden were observed. In this group, a significantly higher proportion of patients with HPV-related OPC was noted. So, in HPV-positive OPC patients, despite the low-intermediate risk (T < 3, N2), the feeding tube placement is observed more frequently, and there was no good adherence of the above-mentioned risk stratification to the prophylactic feeding tube recommendations in this patients group. In the authors’ opinion, the poor nutritional outcomes in HPV (+) OPC patients are associated with a lack of adherence to nutrition recommendations given, and thus suboptimal feeding tube utilization in this patient group [63,64].

Another possible explanation for poorer nutrition outcomes in HPV-related HNC patients is the theory that it could be caused by the increased acute toxicities presented during CRT. Becker-Schiebe et al. [65] reported this phenomenon. Seventy nine HNC patients undergoing CRT or RT-antibody therapy were retrospectively analyzed. In this study, p16 overexpression was detected in 32 patients. Pretreatment anemia was reported in one-third of patients. Only in 5% of patients, both pre-RT anemia and p16 overexpression were noted. The authors reported that p16 expression was significantly associated with acute grade three toxicity. The grade ≥ three radiodermatitis was noted in 47% of p16-positive patients compared to 26% of p16-negative patients (*p* = 0.04). A reduced risk of severe skin toxicities was noted in patients with hypoxic blood values before RT. p16 expression was significantly correlated with local control (*p* = 0.002). The authors concluded that HPV status was associated with better response to RT/CRT but also significantly related to acute high-grade toxicity [64,65].

The mentioned above problems have been widely presented, summarized, and discussed in Brown’s editorial. The author concluded that patients with HPV-associated OPC had higher rates of weight loss and, therefore, they need increased nutritional support [65].

In the other study, Naik et al. [66] analyzed 130 HPV+ and 17 HPV− OPC patients treated exclusively with conventional 3-field RT with chemotherapy. Comparison of the rates of normal diet, limited diet (significant restrictions in the types of foods eaten, and/or requiring nutritional supplementation for weight maintenance), and feeding tube dependence (FTD) between HPV+ and HPV− patients was performed. A median follow-up was 55 months. HPV+ OPC patients more frequently had resumed a normal diet (87% vs. 65%) at last follow up the lower rates of limited diet was noted (9% vs. 18%) and FTD (4% vs. 18%) compared to HPV− patients (*p* = 0.02). On univariate analysis, HPV status was the only significant predictor of reduced swallowing dysfunction (HR 0.19; *p* = 0.008). Moreover, patients treated between 2002 and 2010 had less FTD (7.5% vs. 34%, *p* < 0.001) and dietary limitations (26% vs.46%, *p* < 0.001) at six months post-treatment, compared to patients treated in 1989–2002 group. The authors concluded that HPV+ OPC patients had reduced late swallowing dysfunction following CRT compared to HPV− patients. In the authors’ opinion, the changing epidemiology of OPC (higher incidence of HPV-positive OPC in the currently treated patients) may play a role in toxicity reduction in these patients, independent of the increasing use of intensity-modulated radiotherapy (IMRT) [66].

Vangelov et al. [67] assessed the impact of HPV status on weight loss and feeding tube use in OPC. This retrospective study involved 100 OPC patients undergoing CRT. The feeding tube use and timing of insertion (prophylactic vs. reactive), percentage weight loss during radiotherapy, and the prevalence of critical weight loss (CWL) ≥ 5% in one month were assessed. Prophylactic feeding tubes (PFTs) were inserted prior to or within the first two weeks of radiotherapy and were either a percutaneous endoscopic gastrostomy (PEG) or surgically inserted balloon gastrostomy. Reactive feeding tubes (RFTs) were inserted due to poor oral intake and subsequent weight loss during RT or within two weeks of completion. RFTs were predominantly nasogastric tubes (NGTs) and, in some cases, proceeding to gastrostomy insertion. In HPV-positive patients, higher weight loss during RT compared to the rest of the cohort (8.4% vs. 6.1%, 95%CI 0.8–3.9, *p* = 0.003) was noted. CWL was observed in 86% of patients, and in a significantly higher number of HPV-positive patients (*n* = 63/68, 93%), compared with the rest of group (*n* = 23/32, 72%) (*p* = 0.011). Sixty-one patients required a feeding tube, with the majority inserted reactively (*n* = 33/61, 54%). A larger proportion of HPV-positive patients required feeding tubes (*n* = 43/68, 63% vs. *n* = 18/32, 56% others) (*p* = 0.050), most being RFTs (*n* = 27/43, 63%), and only 4/12 patients presenting with ≥5% loss at diagnosis actually had a PFT inserted. No HPV-positive patients had a therapeutic tube. In HPV-positive patients with PFTs, significantly higher mean percentage weight loss compared to those with PFTs in the remaining cohort (8.6% vs. 3.9% respectively) (mean difference 4.7%, 95% CI 1.7–7.6, *p* = 0.003) was noted. Conditional probability modeling analysis showed, with 74% accuracy, that concurrent CRT and HPV-positive status were predictors of CWL comparing HPV-positive patients to HPV-negative (96%, *p* = 0.001 and 98%, *p* = 0.012 respectively). More HPV-positive patients required feeding tubes (*n* = 43, 63%, *p* = 0.05), most being reactive (*n* = 27, 63%). All patients with reactive tubes experienced CWL. The authors emphasized the high incidence of CWL in patients with HPV-positive OPC and the necessity of nutritional support in this population. In the authors’ opinion, predicting who will require a tube is challenging, and larger, prospective cohort studies are required [67].

HPV positive status in HNC patients undergoing RT/CRT is associated with a higher risk of malnutrition. It is associated with a lack of adherence to nutrition recommendations given and more common acute high-grade toxicity in this patient group. Therefore, in our opinion, these patients should be very carefully qualified for more frequent and earlier nutritional support compared to HPV-negative patients.

## 10. Immunomodulating Nutrition (IN) in HNC Patients Undergoing CRT

It is known that levels of inflammatory, angiogenic, and oxidative stress markers are increased in advanced HNC (stage III/IV). It is also known that oxidative stress has been demonstrated in head and neck cancers. Increased levels of lipid peroxidation products such as lipid peroxide, malondialdehyde and nitric oxide metabolites including nitrite, nitrate, total nitrite and decreased levels of enzymatic and non-enzymatic antioxidants such as superoxide dismutase, glutathione peroxidase, catalase, ascorbate, and alfa-tocopherol were reported in HNC patients [68]. Therefore, it is important to note the influence of the levels of mentioned above markers in HNC patients or modulate immune status in these patients. It has been proven that nutritional support influences systemic inflammatory response [68]. The are non-numerous reports regarding the use of immunomodulating nutrition (immunonutrition, IN) in HNC patients undergoing CRT in the world literature. Immunomodulating nutrients are as follows: amino acids (glutamine, arginine), omega-3 fatty acids, ribonucleic acids. Vitamins and antioxidants are also immunonutriens, according to some authors. 

Arginine is the most common immunonutrient given to HNC patients. It is a non-essential amino acid playing a role in the synthesis of nucleotides, polyamines, nitric oxide, and proline. Arginine stimulates lymphocyte function and improves wound healing. Particularly surgical HNC patients have a high risk of postoperative complications such as wound infections, fistula formation, pneumonia, and sepsis. Moreover, many HNC patients are malnourished due to mechanical obstruction, tumor-induced cachexia, poor dietary habits, and alcohol consumption. The immune defects, including T-lymphocytopenia and dysfunction, decreased monocyte HLA-DR expression, reduced prostaglandin concentration, and antigen-antibody complexes, are reported in this patient group [69]. 

Glutamine is a common nitrogen donor for healing tissues. There are many studies showing benefits of oral or enteral glutamine supplementation in order to improve the quality of life of cancer patients that is associated with better nutrition but also decreased mucosal damage (mucositis, stomatitis, pharyngitis, esophagitis, and enteritis) [70,71]. Additionally, it is a precursor for protein, nucleotide, and nucleic acid synthesis and regulates various cellular pathways and related functions. Therefore, glutamine is important for intestinal integrity and function, proper immunologic response, and antioxidative balance. It is an obligatory nutrient, and its concentration is decreased in patients in catabolism and stress. Glutamine influences cellular activity as a precursor of purine and pyrimidine compounds and a glutathione precursor. It impacts nitric oxide metabolism in an interaction with the abovementioned arginine, regulates cell maturation, stimulates the production of heat shock proteins (HSP), increases the cytotoxicity of tumor necrosis factor (TNF) alpha, and activates kinases responsible for extracellular communication. Additionally, it influences lymphocyte activity, such as stimulation of Con-A and PHA-induced proliferation, activation of CD 25, CD 71, CD 45RO expression, stimulation of Interferon (INF)—gamma secretion, stimulation of natural killer (NK) cells, inhibition of apoptosis, stimulation of GALT, and an increase in NK population in the spleen. Also, it influences the activity of monocytes: stimulation of RNA synthesis, an increase of IL-1 secretion, stimulation of phagocytosis, stimulation of antigen presentation, stimulation of monocyte maturation [71,72].

Omega-3 fatty acids increase the production of some prostaglandins (PGs) and leukotrienes, reducing the proinflammatory potential, and decrease the production of some other PGs (PGE2) and leukotrienes, reducing the cytotoxicity of macrophages, lymphocytes, and natural killer (NK) cells. They decrease prostacyclin and thromboxane (TX)-A2 production and increase the antiaggregatory substance TXA3. It has been reported that omega-3 fatty acids decrease excessive inflammatory responses, but they are not immunosuppressive, which is very important in cancer patients in the postoperative course. Nucleotides are necessary for the proliferation of immune cells and cells important for wound healing [71,72,73,74,75].

Machon et al. [68] analyzed 31 HNC patients undergoing CRT. Immunomodulating nutrition (Oral Impact) was administered five days before each cycle of chemotherapy. At baseline, median levels of inflammatory (C-reactive protein (CRP) 9.8 mg/l (0.8–130.1), Interleukin-6 (Il-6) 4.2 pg/mL (0.7–126.5)), pro-angiogenic (vascular endothelial growth factor( VEGF) 229.5 pg/mL (13.1–595.9)) and pro-oxidative stress (urinary isoprostanes 118 pmol/mmol creatinine (51–299)) markers were increased. CRP (*p* = 0.002) and α-1 acid glycoprotein (*p* = 0.020) levels decreased after five days of IN. Authors concluded that patients with stage III or IV HNC are characterized by a pro-inflammatory, pro-angiogenic, and pro-oxidative status, and therefore, nutritional support could improve this inflammatory status and could prevent severe acute mucositis [66].

In the other study, Zheng et al. [76] conducted a systematic review in order to assess the impact of immunonutrition on CRT patients. In this study, 1478 patients and 27 studies were included. There were no significant differences in the incidence of oral mucositis (relative risk (RR) = 0.91; 95% confidence interval (CI), 0.79–1.05), diarrhea (RR = 0.89; 95% CI, 0.76–1.05), or esophagitis (RR = 0.55; 95% CI, 0.11–2.86) between the IN group and standard nutrition/placebo group. The authors observed that IN significantly reduced the incidence of grade ≥ three oral mucositis (RR = 0.45; 95% CI, 0.22–0.92), grade ≥ three diarrhea (RR = 0.56; 95% CI, 0.35–0.88), grade ≥ three esophagitis (RR = 0.15; 95% CI, 0.04–0.54), and losing > 5% body weight (RR = 0.34; 95% CI, 0.18–0.64). This study showed that immunonutrition failed to reduce the incidence rates of oral mucositis, diarrhea, or esophagitis but was associated with significantly improving the severity of oral mucositis and diarrhea esophagitis and reducing the rate of weight loss [76].

Chao et al. [77] retrospectively reviewed 88 patients with head and neck (HN) and esophageal cancer undergoing CRT. The authors compared two groups: immune-modulating enteral nutrition (IEN) (Impact formula) or standard enteral nutrition (SEN) (isonitrogenous and isoenergetic formula). In approximately 45% of patients, moderate to severe malnutrition was noted (nutritional risk index (NRI) < 97.5) at the beginning of the RCT in the SEN (*n* = 19) and IEN (*n* = 21) groups alike. Significant improvement was observed in the NRI of malnourished patients of the IEN group (97.3 ± 11.9 vs. 98.0 ± 12.0, *p* = 0.021). Additionally, there was a significant difference in the body weight (BW) between the two analyzed groups: BW increased (65.4 ± 14.8 kg vs. 66.3 ± 14.3 kg, *p* = 0.03) in the IEM group and decreased (62.3 ± 12.3 kg vs. 61.7 ± 12.0 kg, *p* = 0.023) in the SEM group. The authors concluded that IN significantly prevented deterioration of nutritional status during CRT compared to SEN [77].

Patients with stage III or IV HNC are characterized by a pro-inflammatory, pro-angiogenic, and pro-oxidative status, and therefore nutritional support could improve the altered inflammatory status and could prevent severe acute mucositis in this patient group. Immunomodulating nutrients (amino acids (glutamine, arginine), omega-3 fatty acids, ribonucleic acids) can be administered in HNC patients undergoing RT/CRT. The most frequently, oral formulas (such as Oral Impact) are used. It has been reported that immunomodulating enteral nutrition (IEN) (Impact formula) significantly prevented deterioration of nutritional status during CRT compared to standard enteral nutrition. Additionally, IN is associated with significantly improving the severity of oral mucositis and diarrhea esophagitis and reducing the rate of weight loss in HNC patients. The reports on the role of IN in HNC patients are contradictory. According to other authors, IN does not influence inflammatory response in HNC patients [68,69]. It should be added that there are not many publications regarding the role of immunonutrition in HNC patients undergoing CRT. The use of immunonutrition is more frequently reported in surgical HNC patients. Therefore, we recommend considering the use of IN in HNC patients undergoing RT/CRT, individually in each patient.

## 11. Malnutrition in Surgical Patients with HNC

Currently, surgery, radiation therapy, and chemotherapy are used in the treatment of HNC. The aim of this paper was to present and discuss nutritional support in HNC patients undergoing CRT. But, surgical HNC patients are also at a high risk of malnutrition. Primary curative surgery is indicated for patients with resectable tumors in which clear margins can be achieved, and function is preserved. Classic open surgery or minimally invasive procedures such as transoral robotic surgery (TORS) or laser surgery are used depending on the anatomy and tumor characteristics. Currently, TORS, with or without neck dissection, is offered as an alternative to CRT in selected patients, particularly with early HNC [78]. Furthermore, in poorer countries, radiotherapy is typically not widely available; as such, radiotherapy is mostly replaced there by far-going surgery. Surgery can lead to additional complications that may result in malnutrition in HNC patients. Enhanced recovery after surgery is used in the nutritional support of patients undergoing surgery for HNC. Inadequate oral intake for more than 14 days is associated with higher mortality. Patients with severe nutritional risk should receive nutrition support for 10–14 days prior to major surgery even if surgery has to be delayed [79,80]. Enteral nutrition is indicated even in patients without obvious undernutrition if it is anticipated that patients will be unable to eat for more than seven days peri-operatively. Early post-operative tube feeding (within 24 h) is indicated in patients in whom early oral nutrition cannot be initiated. Additionally, early oral nutrition should be considered in patients following laryngectomy. In patients with postoperative chyle leak, nutrition with fat-free or high medium-chain triglyceride (MCT) nutritional supplements, either orally or via a feeding tube, is recommended [79,80]. Patients undergoing surgery are exposed to various postoperative complications such as infectious complications (wound infection and wound dehiscence, abscess), fistula, hematoma, seroma [81]. In patients undergoing surgery, particularly with postoperative complications, oral nutrition can be delayed. Therefore, they frequently need enteral nutritional support. Immunonutrition is frequently used in this patient group [81,82]. It has been proven that IN is associated with shorter hospital stay and a lower rate of wound infections and local complications in HNC patients undergoing surgery [82].

## 12. Summary

In summary, it is certain and indisputable that nutritional support is necessary in all HNC patients undergoing CRT in order to improve nutritional status and prevent malnutrition caused by the adverse effect of oncological treatment. Therefore, nutritional intervention is needed both in malnourished (to improve nutritional status) and well-nourished (to prevent malnutrition) patients at the beginning of treatment, nutritional counseling with ONS should be introduced. When the energy and protein intake through the oral route is not sufficient, EN should be started. The choice of optimal timing of introduction of enteral nutrition and kind of feeding tube is challenging. Decisions should be made by a multidisciplinary team. There are contradictory reports regarding the benefits and disadvantages of prophylactic and reactive FT. It is important to introduce EN not too late for patients, but also not too early, in order not to expose the patient to the risk of unnecessary complications associated with FT placement. On the other side, the choice of optimal route for EN (GNT vs. PEG) is also challenging because both methods have advantages and disadvantages. Therefore, many authors in their study try to create an algorithm of nutritional support in HNC patients undergoing CRT. This algorithm is very important and necessary. Therefore, multi-center prospective randomized control trials are needed in order to precise the indications for the use of different EN methods in HNC patients undergoing CRT. Also, the optimal timing of nutritional intervention is important. Most authors recommend to start nutritional support before CRT and continue it during and after CRT because deterioration of nutritional status is observed in HNC patients for a very long time after CRT (to 12 months). Also, a separate algorithm of nutritional support for HPV-related OPC patients is needed because it was shown that this patient group is at high risk for malnutrition. HPV-related OPC is another HNC type not only considering etiology, epidemiology, and de-escalation of treatment doses, but also a higher malnutrition risk and a higher toxicity rate in patients following CRT. Immunonutrition can improve deteriorated immune status in patients with advanced HNC patients undergoing CRT, but there are not many reports regarding the use of IN in this patients group. Also, further investigations on the use of IN in HNC patients undergoing CRT are needed. Prospective studies finding the precise indications for enteral nutrition should be performed. Also, the timing of insertion of nasogastric tubes or PEGs should be precise. The optimal timing and precise indications for nutritional support are the most important. In our opinion, a separate qualification algorithm for nutritional support should be created for patients with HPV-related head and neck cancers. Currently, nutritional support should be planned and used for each patient individually, considering ESPEN guidelines for EN and previous publication on this topic.

**Table 3 nutrients-13-00057-t003:** Summary of Studies on Nutritional Support in Head and Neck Cancer Patients Undergoing Chemoradiotherapy.

Reference (Year)	Patients No.	Study Design	Nutritional Intervention	Outcome
Hu et al. [25] (2020)	243 HNC	Retrospective	Nutritional counseling (timing)	Weight, early CT termination or incompletion
Jovanovic et al. [26] (2020)	122 OPC	Retrospective	Feeding tube	Weight, swallowing, quality of life
Paccagenella et al. [27] (2010)	66 HNC	Retrospective	Early nutrition before CRT vs. control	Weight, RT interruptions/delay, unplanned hospitalization
Fraser et al. [28] (2020)	250 HNC	Retrospective	Prophylactic PEG	A number of days Nil by Mouth (NBM)
Vlooswijk et al. [29] (2016)	276 OPC	Retrospective	Dietary counseling	Weight, toxity (dysphagia, mucositis, toxity grade), OS, DFS
Kabarriti et al. [30] (2018)	352 HNC	Retrospective	Dietary counseling	BMI, sarcopenia, OS
Valentini et al. [31] (2012)	21 HNC	Prospective	Dietary counseling, ONS	BMI, mid arm circumference, mucositis grade
Mello et al. [32] (2020)	15 trials HNC	Meta-analysis	Dietary counseling, ONS	Mortality, tolerance, QoL, FS, adverse effects
Bicakli et al. [33] (2017)	69 HNC	Prospective	Dietary counseling	Weight, body composition parameters, mucositis
Van den Berg et al. [34] (2010)	38 HNC	Prospective	Dietary counseling	Weight, BMI, malnutrition
Ferreira et al. [35] (2020)	65 HNC	Prospective	Dietary counseling, ONS	Weight, energy, macro- and micronutrients intake
Cereda et al. [36] (2017)	159 HNC	Prospective, R	Dietary counseling, ONS	Weight, protein-calorie intake, muscle strength, QoL
Ravasco et al. [37] (2005)	75 HNC	Prospective, R	Dietary counseling, ONS	Energy-protein intake, OoL, adverse effects
Langius et al. [38] (2013)	12 reports HNC	Systematic review	Dietary counseling, ONS, NS, PEG	QoL
Vangelov et al. [39] (2017)	131 OPC	Retrospective	Reactive FT vs. prophylactic FT	Weight, Survival
Cady [40] (2007)	Review	Review	PEG	Patient-, tumor-, treatment-related factors
Lang et al. [41] (2020)	186 HNC	Retrospective	PEG	BMI, nutritional intake, adverse effects
Brown et al. [42] (2018)	130 HNC	Prospective	Prophylactic PEG vs. no-PEG	Weight, unplanned hospital admission
Brown et al. [43] (2017)	70 HNC	Prospective, R	Prophylactic PEG vs. no-PEG	Weight, QoL
Zhang et al. [44] (2016)	1631 HNC	Meta-analysis	pPEG/NGT vs. rPEG/NGT	Weight, treatment interruption, hospital admission
Kramer et al. [45] (2014)	74 HNC	Historical cohort	pPEG vs. rPEG	Weight, survival, disease control
Willemsen et al. [46] (2020)	450 HNC	Retrospective	Prophylactic PEG	PEG-dependency
Van der Linden et al. [47] (2017)	240 HNC	Retrospective	PEG, NGT	Lifestyle, oncological, treatment, nutrition outcome
Sieron et al. [48] (2020)	212 HNC	Retrospective	Prophylactic PEG/RPG	Weight, BMI, tube-related complications
Haussman et al. [49] (2019)	181 HNC	Retrospective	Prophylactic PEG	NRS, tube-related complications
Yanni et al. [50] (2019)	152 HNC	Retrospective	pPEG vs. rPEG	BMI, readmission, dysphagia, health, complications
McClelland et al. [51] (2018)	25 reports	Systematic review	pPEG vs. rPEG	Weight, QoL, PEG-dependence
Strom et al. [52] (2013)	297 OPC	Retrospective	Reactive PEG	BMI, oncological and treatment factors for rPEG placement
Silander et al. [53] (2012)	134 HNC	Prospective	pPEG vs. control group	Duration of hospitalization, QoL
Assenat et al. [54] (2011)	139 HNC	Retrospective	pPEG vs. No-pPEG	Duration of hospitalization, treatment interruption, NS
O’Shea et al. [55] (2015)	104 HNC	Retrospective	Prophylactic PEG	Smoking, alcohol consumption, PEG duration/dependence
Sadasivan et al. [56] (2012)	XX HNC	Prospective	PEG vs. NGT	QoL, tube-related complications
Corry et al. [57] (2009)	105 HNC	Prospective	PEG vs. NGT	Weight, QoL, cost, dysphagia, tube duration, complications
Mekhail et al. [58] (2001)	158 HNC	Retrospective	PEG vs. NGT	Dysphagia, tube duration
Nugent et al. [12] (2013)	Review HNC	Review	PEG vs. NGT	Weight, anthropometry, tube duration, cost
Magne et al. [59] (2001)	90 HNC	Prospective	NGT vs. PFG	QoL, weight, BMI, tube—failure, duration, complications
Wiliams et al. [60] (2012)	104 OPC	Retrospective	pPEG vs. rPEG vs. NGT	Weight, tube duration, diet type
Ward et al. [61] (2016)	78 LPC	Retrospective	rNGT vs. pPEG vs. rPEG	Severe late dysphagia
Anderson et al. [62] (2018)	139 HNC	Retrospective	Feeding tube	Risk factors (tumor, nodal staging) for FT placement
Anderson et al. [63] (2019)	101 OPC	Retrospective	Feeding tube, HPV status	Risk factors (tumor, nodal staging, HPV) for FT placement
Vangelov et al. [67] (2018)	100 OPC	Retrospective	pFT vs. rFT, HPV status	Critical weight loss (CWL), tube timing depending on HPV
Machon et al. [68] (2012)	31 HNC	Clinical trial	IN (Oral Impact) before CRT	Acute-phase proteins, cytokines before and after IN
Zheng et al. [76] (2020)	1487 CRT	Meta-analysis	IN vs. standard/placebo group	Oral mucositis, diarrhea, esophagitis
Chao et al. [77] (2020)	88 HNC	Retrospective	IN (Impact) vs. standard group	Weight, NRI

No., number; OS, overall survival; DFS, disease-free survival; BMI, body mass index; ONS, oral nutritional supplements; FS, functional status; R, randomized; NS, nasogastric tube, PEG, percutaneous endoscopic gastrostomy; FT, feeding tube; PFT; p, prophylactic; r, reactive; NGT, nasogastric tube; RPG, radiological percutaneous gastrostomy; NRS, nutritional risk score; NS, nutritional status; PFG, percutaneous fluoroscopic gastrostomy; LPC, laryngopharynx cancer; IN, immunonutrition; NRI, nutritional risk index; QoL: Quality of life.

## Figures and Tables

**Table 1 nutrients-13-00057-t001:** Dysphagia in Head and Neck Cancer Patients: Etiopathology.

Dysphagia Type	Cause/Biomolecular Lesions	Pathology/Clinical Manifestation
Tumor-related dysphagia	Compression by the tumor	Obstruction by the tumor volume
	Neoplastic infiltration	Infiltration of structures involved in swallowing
Surgery-related dysphagia	Radical oncological resection	Resection of structures involved in swallowing
	Postoperative complications	Postoperative inflammation, edema compressing, or
	(seroma, hematoma, abscess, fistula, wound infection/dehiscence, and chyle leak)	infiltrating structures involved with swallowing
Radiochemotherapy-related dysphagia		
Early dysphagia	Reactive oxygen/nitrogen species	Edema, erythema, leukocyte infiltration, vasodilatation,
	Impaired cell proliferation	Vascular leakage, hypoplasia
	Epithelial denudation	Xerostomia, dysgeusia, mucositis, and inflammation
Interval dysphagia	Oxidative injury/genetic changes	Increased fibroblast growth, increased collagen content
Delayed dysphagia	Currently unknown	Persistent disorganized matrix, vascular changes, Fibrosis, and atrophia

**Table 2 nutrients-13-00057-t002:** Strategy of the Literature Review.

1. All papers on “head and neck cancer” or “radiotherapy” or “nutrition” in the PubMed (813,289 results) and Web of Science (559,479) database were found.
2. The articles on nutrition in head and neck cancer patients undergoing chemoradiotherapy (352 articles in PubMed; 466 articles in Web of Science) were chosen and reviewed.
3. Papers regarding various forms of nutritional support (dietary counseling, oral nutritional supplements, enteral feeding including prophylactic vs. reactive enteral nutrition and nasogastric tube vs. percutaneous gastrostomy, immunonutrition) and human papillomavirus status (74 articles) were selected and discussed.
4. Additionally, articles on characteristics of immunonutrients (glutamine, arginine, omega-3 fatty acids, and nucleotides), in order to show the role of their role in cancer patients, were found, reviewed, and cited.

## Data Availability

Not applicable.

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
