# Peer review of "Nutritional Support in Head and Neck Radiotherapy Patients Considering HPV Status"

_nutrients, 2020, doi:10.3390/nu13010057_

Round 1
Reviewer 1 Report
The authors provide a rather comprehensive review concerning malnutrition in HNC patients, with an emphasis (in part) on HPV-associated oropharyngeal cancers. I do have some comments:
- Although the article mainly concerns radio- and chemotherapy, cancers at an early stage typically require mainly surgery, but malnutrition can already be fairly abundant in this group of patients, and the authors should add a small paragraph discussing this feature.
- In poorer countries, radiotherapy is typically not widely available; as such, radiotherapy is mostly replaced there by far-going surgery, causing other forms of malnutrition, and the authors should also add a small paragraph discussing this.
- While immunomodulation using nutrition has a high potential in immunity-associated types of cancers (such as those caused by HPV), the quality of evidence is not straightforward, and the authors should be cautious using strong conclusions.
- In their Summary, the authors should note more precisely what needs to be done next, in order to obtain an algorithm for using nutritional support in HNC patients.
- The word "conclusion" can be omitted in each paragraph
- In paragraph 7, the numbering is not correct.
- The English is substandard and should be improved.
Author Response
Dear Editor,
Thank you for peer reviewing of our manuscript nutrients- 1046651, entitled " Nutritional Support in Head and Neck Radiotherapy Patients Considering HPV Status".
Thank you for your questions and comments. We have fully addressed all the comments and my responses appear below. Our revised work includes corrections according to reviewers’ comments in the text. The changes, made according to reviewers’ comments, are highlighted in red print in the text.
We take this opportunity to express my gratitude to the reviewers for their constructive and useful remarks. Their comments allowed us to identify areas in my manuscript that needed modification.
We also thank you for allowing me to resubmit a revised copy of the manuscript.
We hope that the revised manuscript is now acceptable for publication in Nutrients.
Responses to Reviewer 1.
- Although the article mainly concerns radio- and chemotherapy, cancers at an early stage typically require mainly surgery, but malnutrition can already be fairly abundant in this group of patients, and the authors should add a small paragraph discussing this feature.
Answer:
A small paragraph regarding the role of surgery in head and neck cancer (HNC) patients was added. Additionally, causes of surgery-related dysphagia were presented in Table (below the references in the manuscript)1. In this paragraph, the causes of malnutrition in surgical patients were presented, as follows:
- Malnutrition in surgical patients with HNC
Currently, surgery, radiation therapy and chemotherapy are used in the treatment of HNC. The aim of this paper was to present and discuss nutritional support in HNC patients undergoing CRT. But, surgical HNC patients are also in the high risk of malnutrition. Primary curative surgery is indicated for patients with resectable tumors in which clear margins can be achieved and function is preserved. Classic open surgery or minimally invasive procedures such as transoral robotic surgery (TORS) or laser surgery are used depending on the anatomy and tumor characteristics. Currently, TORS, with or without neck dissection, is offered as an alternative to CRT in selected patients, particularly with early HNC [78]. Furthermore, in poorer countries, radiotherapy is typically not widely available; as such, radiotherapy is mostly replaced there by far-going surgery. Surgery can lead to additional complications that may result in malnutrition in HNC patients. Enhanced recovery after surgery are used in the nutritional support in patients undergoing surgery for HNC. Inadequate oral intake for more than 14 days is associated with a higher mortality. Patients with severe nutritional risk should receive nutrition support for 10–14 days prior to major surgery even if surgery has to be delayed [79,80]. Enteral nutrition is indicated even in patients without obvious undernutrition, if it is anticipated that patients will be unable to eat for more than 7 days peri-operatively. Early post-operative tube feeding (within 24 hours) is indicated in patients in whom early oral nutrition cannot be initiated. Additionally, early oral nutrition should be considered in patients following laryngectomy. In patients with postoperative chyle leak, nutrition with fat free or high (medium chain triglyceride) MCT nutritional supplements either orally or via a feeding tube is recommended [79,80]. Patients undergoing surgery are exposed to various postoperative complications such as infectious complications (wound infection and wound dehiscence, abscess), fistula, hematoma, seroma [81]. In patients undergoing surgery, particularly, with postoperative complications, oral nutrition can be delayed. Therefore, they frequently need enteral nutritional support. Immunonutrition is frequently used in this patients group [81,82]. It has been proven that IN is associated with shorter hospital stay and lower rate of wound infections and local complications in HNC patients undergoing surgery [82].
Table 1 Dysphagia in HNC patients: etiopathology (in the manuscript).
- In poorer countries, radiotherapy is typically not widely available; as such, radiotherapy is mostly replaced there by far-going surgery, causing other forms of malnutrition, and the authors should also add a small paragraph discussing this.
Answer:
A small paragraph regarding the role of surgery in head and neck cancer (HNC) patients was added. Also, the causes of dysphagia in surgical patients are presented in mentioned above Table 1. In this paragraph, the causes of malnutrition in surgical patients were presented, as follows:
Furthermore, in poorer countries, radiotherapy is typically not widely available; as such, radiotherapy is mostly replaced there by far-going surgery. Surgery can lead to additional complications that may result in malnutrition in HNC patients. Enhanced recovery after surgery are used in the nutritional support in patients undergoing surgery for HNC. Inadequate oral intake for more than 14 days is associated with a higher mortality. Patients with severe nutritional risk should receive nutrition support for 10–14 days prior to major surgery even if surgery has to be delayed [79,80]. Enteral nutrition is indicated even in patients without obvious undernutrition, if it is anticipated that patients will be unable to eat for more than 7 days peri-operatively. Early post-operative tube feeding (within 24 hours) is indicated in patients in whom early oral nutrition cannot be initiated. Additionally, early oral nutrition should be considered in patients following laryngectomy. In patients with postoperative chyle leak, nutrition with fat free or high (medium chain triglyceride) MCT nutritional supplements either orally or via a feeding tube is recommended [79,80]. Patients undergoing surgery are exposed to various postoperative complications such as infectious complications (wound infection and wound dehiscence, abscess), fistula, hematoma, seroma [81]. In patients undergoing surgery, particularly, with postoperative complications, oral nutrition can be delayed. Therefore, they frequently need enteral nutritional support. Immunonutrition is frequently used in this patients group [81,82]. It has been proven that IN is associated with shorter hospital stay and lower rate of wound infections and local complications in HNC patients undergoing surgery [82].
Table 1 Dysphagia in HNC patients: etiopathology (in the manuscript).
- While immunomodulation using nutrition has a high potential in immunity-associated types of cancers (such as those caused by HPV), the quality of evidence is not straightforward, and the authors should be cautious using strong conclusions.
Answer:
It was added that the role of immunonutrition in HNC patients undergoing CRT has been not reported in many publications, the reports on immunonutrition are contradictory and this nutritional suport should be considered individually in the each patient, as follows:
The reports on the role of IN in HNC patients are contradictory. According to other authors, IN does not influence on inflammatory response in HNC patients [68,69]. It should be added that there are not many publications regarding the role of immunonutrition in HNC patients undergoing CRT. The use of immunonutrition is more frequently reported in surgical HNC patients. Therefore, we recommend to consider the use of IN in HNC patients undergoing RT / CRT, individually in each patient.
- In their Summary, the authors should note more precisely what needs to be done next, in order to obtain an algorithm for using nutritional support.
Answer:
The needed studies were more precisely noted, as follows:
Prospective studies finding the precise indications for enteral nutrition should be performed. Also, the timing of insertion of nasogastric tube or PEG should be precise. The optimal timing and precise indications for nutritional support are the most important. In our opinion, a separate qualification algorithm for nutritional support should be created for patients with HPV-related head and neck cancers.
- The word "conclusion" can be omitted in each paragraph.
Answer:
The word "conclusion" was omitted in each paragraph.
- In paragraph 7, the numbering is not correct.
Answer:
The numbering has been corrected (A number of paragraph 7 was changed (paragraph 8), because one paragraph (dysphagia) was added according to 2nd reviewer’s suggestion, as follows:
8.2. Nasogastric tube versus percutaneous endoscopic gastrostomy in HNC patients undergoing CRT
- The English is substandard and should be improved.
Answer:
The English has been improved.
Dear Editor,
Thank you for peer reviewing of our manuscript nutrients- 1046651, entitled " Nutritional Support in Head and Neck Radiotherapy Patients Considering HPV Status".
Thank you for your questions and comments. We have fully addressed all the comments and my responses appear below. Our revised work includes corrections according to reviewers’ comments in the text. The changes, made according to reviewers’ comments, are highlighted in red print in the text.
We take this opportunity to express my gratitude to the reviewers for their constructive and useful remarks. Their comments allowed us to identify areas in my manuscript that needed modification.
We also thank you for allowing me to resubmit a revised copy of the manuscript.
We hope that the revised manuscript is now acceptable for publication in Nutrients.
Responses to Reviewer 1.
- Although the article mainly concerns radio- and chemotherapy, cancers at an early stage typically require mainly surgery, but malnutrition can already be fairly abundant in this group of patients, and the authors should add a small paragraph discussing this feature.
Answer:
A small paragraph regarding the role of surgery in head and neck cancer (HNC) patients was added. Additionally, causes of surgery-related dysphagia were presented in Table (below the references in the manuscript)1. In this paragraph, the causes of malnutrition in surgical patients were presented, as follows:
- Malnutrition in surgical patients with HNC
Currently, surgery, radiation therapy and chemotherapy are used in the treatment of HNC. The aim of this paper was to present and discuss nutritional support in HNC patients undergoing CRT. But, surgical HNC patients are also in the high risk of malnutrition. Primary curative surgery is indicated for patients with resectable tumors in which clear margins can be achieved and function is preserved. Classic open surgery or minimally invasive procedures such as transoral robotic surgery (TORS) or laser surgery are used depending on the anatomy and tumor characteristics. Currently, TORS, with or without neck dissection, is offered as an alternative to CRT in selected patients, particularly with early HNC [78]. Furthermore, in poorer countries, radiotherapy is typically not widely available; as such, radiotherapy is mostly replaced there by far-going surgery. Surgery can lead to additional complications that may result in malnutrition in HNC patients. Enhanced recovery after surgery are used in the nutritional support in patients undergoing surgery for HNC. Inadequate oral intake for more than 14 days is associated with a higher mortality. Patients with severe nutritional risk should receive nutrition support for 10–14 days prior to major surgery even if surgery has to be delayed [79,80]. Enteral nutrition is indicated even in patients without obvious undernutrition, if it is anticipated that patients will be unable to eat for more than 7 days peri-operatively. Early post-operative tube feeding (within 24 hours) is indicated in patients in whom early oral nutrition cannot be initiated. Additionally, early oral nutrition should be considered in patients following laryngectomy. In patients with postoperative chyle leak, nutrition with fat free or high (medium chain triglyceride) MCT nutritional supplements either orally or via a feeding tube is recommended [79,80]. Patients undergoing surgery are exposed to various postoperative complications such as infectious complications (wound infection and wound dehiscence, abscess), fistula, hematoma, seroma [81]. In patients undergoing surgery, particularly, with postoperative complications, oral nutrition can be delayed. Therefore, they frequently need enteral nutritional support. Immunonutrition is frequently used in this patients group [81,82]. It has been proven that IN is associated with shorter hospital stay and lower rate of wound infections and local complications in HNC patients undergoing surgery [82].
Table 1 Dysphagia in HNC patients: etiopathology (in the manuscript).
- In poorer countries, radiotherapy is typically not widely available; as such, radiotherapy is mostly replaced there by far-going surgery, causing other forms of malnutrition, and the authors should also add a small paragraph discussing this.
Answer:
A small paragraph regarding the role of surgery in head and neck cancer (HNC) patients was added. Also, the causes of dysphagia in surgical patients are presented in mentioned above Table 1. In this paragraph, the causes of malnutrition in surgical patients were presented, as follows:
Furthermore, in poorer countries, radiotherapy is typically not widely available; as such, radiotherapy is mostly replaced there by far-going surgery. Surgery can lead to additional complications that may result in malnutrition in HNC patients. Enhanced recovery after surgery are used in the nutritional support in patients undergoing surgery for HNC. Inadequate oral intake for more than 14 days is associated with a higher mortality. Patients with severe nutritional risk should receive nutrition support for 10–14 days prior to major surgery even if surgery has to be delayed [79,80]. Enteral nutrition is indicated even in patients without obvious undernutrition, if it is anticipated that patients will be unable to eat for more than 7 days peri-operatively. Early post-operative tube feeding (within 24 hours) is indicated in patients in whom early oral nutrition cannot be initiated. Additionally, early oral nutrition should be considered in patients following laryngectomy. In patients with postoperative chyle leak, nutrition with fat free or high (medium chain triglyceride) MCT nutritional supplements either orally or via a feeding tube is recommended [79,80]. Patients undergoing surgery are exposed to various postoperative complications such as infectious complications (wound infection and wound dehiscence, abscess), fistula, hematoma, seroma [81]. In patients undergoing surgery, particularly, with postoperative complications, oral nutrition can be delayed. Therefore, they frequently need enteral nutritional support. Immunonutrition is frequently used in this patients group [81,82]. It has been proven that IN is associated with shorter hospital stay and lower rate of wound infections and local complications in HNC patients undergoing surgery [82].
Table 1 Dysphagia in HNC patients: etiopathology (in the manuscript).
- While immunomodulation using nutrition has a high potential in immunity-associated types of cancers (such as those caused by HPV), the quality of evidence is not straightforward, and the authors should be cautious using strong conclusions.
Answer:
It was added that the role of immunonutrition in HNC patients undergoing CRT has been not reported in many publications, the reports on immunonutrition are contradictory and this nutritional suport should be considered individually in the each patient, as follows:
The reports on the role of IN in HNC patients are contradictory. According to other authors, IN does not influence on inflammatory response in HNC patients [68,69]. It should be added that there are not many publications regarding the role of immunonutrition in HNC patients undergoing CRT. The use of immunonutrition is more frequently reported in surgical HNC patients. Therefore, we recommend to consider the use of IN in HNC patients undergoing RT / CRT, individually in each patient.
- In their Summary, the authors should note more precisely what needs to be done next, in order to obtain an algorithm for using nutritional support.
Answer:
The needed studies were more precisely noted, as follows:
Prospective studies finding the precise indications for enteral nutrition should be performed. Also, the timing of insertion of nasogastric tube or PEG should be precise. The optimal timing and precise indications for nutritional support are the most important. In our opinion, a separate qualification algorithm for nutritional support should be created for patients with HPV-related head and neck cancers.
- The word "conclusion" can be omitted in each paragraph.
Answer:
The word "conclusion" was omitted in each paragraph.
- In paragraph 7, the numbering is not correct.
Answer:
The numbering has been corrected (A number of paragraph 7 was changed (paragraph 8), because one paragraph (dysphagia) was added according to 2nd reviewer’s suggestion, as follows:
8.2. Nasogastric tube versus percutaneous endoscopic gastrostomy in HNC patients undergoing CRT
- The English is substandard and should be improved.
Answer:
The English has been improved.

Reviewer 2 Report
This is an interesting review about nutritional support in head and neck radiotherapy patients considering HPV status. The authors also reported ESPEN guidelines.
Since the authors performed a systematic review of PubMed and Web of Science, they should report results according to PRISMA guidelines.
I think that a greater explanation of the reasons for dysphagia because of disease and treatments must be added.
Results can be better exposed through tables including data about studies in literature, instead of long description in the text. The latter should only resume the findings and discuss them. The text is too long and difficult to read.
More details about mechanisms of action of immunonutrition may be helpful.
There are several grammatical errors. Too much acronyms are present in the text. The authors should submit the text to a native English speaker to improve readability.
Author Response
Dear Editor,
Thank you for peer reviewing of our manuscript nutrients- 1046651, entitled " Nutritional Support in Head and Neck Radiotherapy Patients Considering HPV Status".
Thank you for your questions and comments. We have fully addressed all the comments and my responses appear below. Our revised work includes corrections according to reviewers’ comments in the text. The changes, made according to reviewers’ comments, are highlighted in red print in the text.
We take this opportunity to express my gratitude to the reviewers for their constructive and useful remarks. Their comments allowed us to identify areas in my manuscript that needed modification.
We also thank you for allowing me to resubmit a revised copy of the manuscript.
We hope that the revised manuscript is now acceptable for publication in Nutrients.
Responses to Reviewer 2.
- Since the authors performed a systematic review of PubMed and Web of Science, they should report results according to PRISMA guidelines.
Answer:
PRISMA guidelines were used: Strategy of the literature review was presented in Table 2, and criteria of seletion of analyzed articles were added.
Full-text articles published in English were included to our review if they met the following criteria:
- Population: Head and neck cancer patients receiving radiotherapy or chemoradiotherapy, including HPV-related cancer patients.
- Interventions: nutritional counselling, oral nutritional supplements, enteral feeding including prophylactic vs. reactive nutrition and nasogastric tube vs. percutaneous endoscopic gastrostomy, immunonutrition.
- Outcomes of interest: Optimal timing of nutritional support; dietary counselling (timing and indications), oral nutritional supplements (timing and indications); comparison of prophylactic and reactive enteral feeding; comparison of nasogastric tube and percutaneous endoscopic gastrostomy, including following parameters: difference in body weight, rates of treatment interruption, nutrition-related hospital admission, and tube-related complications; influence of HPV status on malnutrition and nutritional support; use of immunonutrition in HNC patients undergoing CRT.
Strategy of the literature review is presented in Table 2. The search results are presented in Table 3.
Our article is a comprehensive review on nutrition support in head and neck cancer patients, It is not meta-analysis. Therefore, there is no statistical analysis, but comprehensive discussion, of the search results. The results were summarized in Table 3.
Table 2 Strategy of the Literature Review.
Table 3 Summary of Studies on Nutritional Support in Head and Neck Cancer Patients Undergoing Chemoradiotherapy.
- I think that a greater explanation of the reasons for dysphagia because of disease and treatments must be added.
Answer:
The reasons for dysphagia because of disease and treatments were added in the paragraph 2 and Table 1 (in the manuscript below the references).
- Dysphagia in HNC patients: reasons and treatment
It should be noted that in HNC patients, dysphagia can appear before any oncological treatment and (secondary to the basic disease) and during or after oncological treatment (surgery, chemoradiotherapy) as the result of the treatment toxity.
Dysphagia, defined as the difficulty in swallowing liquids, food, or medication, can occur during the oropharyngeal or the esophageal phase of swallowing. Swallowing dysfunction has been reported in 30–50% of non-surgically treated HNC. Before radiotherapy, dysphagia is caused by obstruction by the tumor volume or infiltration of structures involved with swallowing. In surgical patients, resection of structures necessary for normal deglutition leads to a swallowing dysfunction. In patients receiving radiotherapy, dysphagia is secondary to injury of neural and soft tissues. RT-induced swallowing dysfunction may occur both during treatment and as a late effect of therapy. Acute dysphagia is generally associated with soft tissue inflammation, edema, pain, mucous production, and xerostomia. After radiation has completed, soft tissues are able to heal. In some patients, the healing process results in soft tissue fibrosis, lymphedema, scar tissue formation, and neurological impairment. This also may lead to a swallowing dysfunction [21].
At the beginning, preventive swallowing dysfunction evaluation is used by nutritionists and deglutologists. Usually deglutologists identify swallowing abnormalities, prescribe additional testing (clinical/radiological tests) in order to assess inhalation/aspiration risks, and develop an appropriate treatment plan (correction of swallowing mechanisms through patient education and exercises). Two types of exercises are recommended for patients with dysphagia. They can be performed at the beginning, during and after treatment: indirect (exercises to strengthen swallowing muscles) and direct (postural exercises to be performed while swallowing). The aim of swallowing exercises is to increase the range of movement of the tongue, lips, and jaw. Exercising swallowing muscles improves and/or maintain the possibility of swallowing. The nutritional support involving artificial nutrition is the next step of dysphagia treatment. First of all it includes dietary counselling, oral nutritional supplements and enteral nutrition. If enteral nutrition is introduced, patients should be encouraged to continue to swallow and to wean from artificial nutrition as quickly and safely as is feasible, regardless of the nutrition method (e.g. nasogastric tube, PEG, and parenteral nutrition) [21]. Depending on timing, dysphagia is divided into: early (onset less than 6 months), interval (onset from 6 months to 5-10 years) and late (onset more than 6 months). The pathogenesis of each above mentioned dysphagia type is different [22]. The causes of dysphagia in HNC patients are summarized in Table1.
Table 1 Dysphagia in Head and Neck Cancer Patients: Etiopathology (in the manuscript).
- Results can be better exposed through tables including data about studies in literature, instead of long description in the text. The latter should only resume the findings and discuss them. The text is too long and difficult to read.
Answer:
The table including data about studies in literature was added (Table 3 in the manuscript). The text includes only the most important findings of the presented studies (which can not be presented in a clear form in a table, with such a large amount of research) and it is not possible to remove these results from the text, because it would be associated with a lack of essential informations.
Table 3 Summary of Studies on Nutritional Support in Head and Neck Cancer Patients Undergoing Chemoradiotherapy.
- More details about mechanisms of action of immunonutrition may be helpful.
Answer:
More details about mechanisms of action of immunonutrition were presented.
It is known that levels of inflammatory, angiogenic and oxidative stress markers are increased in advanced HNC (stage III / IV). It is also known that oxidative stress has been demonstrated in head and neck cancers. Increased levels of lipid peroxidation products such as lipid peroxide, malondialdehyde and nitric oxide metabolites including nitrite, nitrate, total nitrite and decreased levels of enzymatic and non-enzymatic antioxidants such as superoxide dismutase, glutathione peroxidase, catalase, ascorbate and alfa-tocopherol were reported in HNC patients [68]. Therefore, it is important to influence on the levels of mentioned above markers in HNC patients or modulate immune status in these patients. It has been proven that nutritional support influenced on systemic inflammatory response [68]. The are non-numerous reports regarding the use of immunomodulating nutrition (immunonutrition, IN) in HNC patients undergoing CRT in the world literature. Immunomodulating nutrients are as follows: amino acids (glutamine, arginine), omega-3 fatty acids, ribonucleic acids. Vitamins, and antioxidants are also immunonutriens according to some authors.
Arginine is the most common immunonutrient given to HNC patients. It is a non-essential amino acid playing a role in the synthesis of nucleotides, polyamines, nitric oxide and proline. Arginine stimulates lymphocyte function and improves wound healing. Particularly surgical HNC patients have a high risk of postoperative complications such as wound infections, fistula formation, pneumonia and sepsis. Moreover, many HNC patients are malnourished due to mechanical obstruction, tumor induced cachexia, poor dietary habits and alcohol consumption. The immune defects, including T-lymhocytopenia and dysfuntion, decreased monocyte HLA-DR expression, reduced prostaglandin concentration and antigen-antibody complexes, are reported in this patients group [69].
Glutamine is a common nitrogen donor for healing tissues. There are many studies showing benefits of oral or enteral glutamine supplementation in order to improve quality of life of cancer patients that is associated with better nutrition, but also decreased mucosal damage (mucositis, stomatitis, pharyngitis, esophagitis, and enteritis) [70,71]. Additionally, it is a precursor for protein, nucleotide, and nucleic acid synthesis and regulates various cellular pathways and related functions. Therefore, glutamine is important for intestinal integrity and function, proper immunologic response and antioxidative balance. It is an obligatory nutrient and its concentration is decreased in patients in catabolism and stress. Glutamine influences on cellular activity: as a precursor of purine and pyrimidine compounds, glutathione precursor. It impacts on nitric oxide metabolism in interaction with mentioned above arginine, regulates cell maturation, stimulates production of hot shot proteins (HSP) increases cytotoxicity of tumor necrosis factor (TNF) alpha, activates kinases responsible for extracellular communication. Additionally, it influences on lymphocyte activity, such as: stimulation of Con-A and PHA-induced proliferation, activation of CD 25, CD 71, CD 45RO expression, stimulation of Interferon (INF) - gamma secretion, stimulation of natural killer (NK) cells, inhibition of apoptosis, stimulation of GALT, increase of NK population in the spleen. Also, it influences on the activity of monocytes: stimulation of RNA synthesis, increase of IL-1 secretion, stimulation of phagocytosis, stimulation of antigen presentation, stimulation of monocyte maturation [71,72].
Omega-3 fatty acids increase production of some prostaglandins (PGs) and leukotrienes, reducing the proinflammatory potential, and decrease production of some other PGs (PGE2) and leukotrienes, reducing the cytotoxicity of macrophages, lymphocytes, and natural killer (NK) cells. They decrease prostacyclin and thromboxane (TX)-A2 production and increase the antiaggregatory substance TXA3. It has been reported that omega-3 fatty acids decrease excessive inflammatory responses, but they are not immunosuppressive, what is very important in cancer patients in the postoperative course. Nucleotides are necessary for proliferation of immune cells and cells important for wound healing [71-75].
- There are several grammatical errors. Too much acronyms are present in the text. The authors should submit the text to a native English speaker to improve readability.
Answer:
The manuscript has been checked and English form and grammar have been improved. The number of acronyms has been decreased.
Round 2
Reviewer 2 Report
Thanks for improving the manuscript.